# The Perlman syndrome DIS3L2 exoribonuclease safeguards endoplasmic reticulum-targeted mRNA translation and calcium ion homeostasis

Mehdi Pirouz [1,2,3], Chih-Hao Wang [4], Qi Liu[1,2], Aref G. Ebrahimi[5], Farnaz Shamsi [4], Yu-Hua Tseng [4,6] & Richard I. Gregory [1,2,6,7,8✉]

DIS3L2-mediated decay (DMD) is a surveillance pathway for certain non-coding RNAs (ncRNAs) including ribosomal RNAs (rRNAs), transfer RNAs (tRNAs), small nuclear RNAs (snRNAs), and RMRP. While mutations in *DIS3L2* are associated with Perlman syndrome, the biological significance of impaired DMD is obscure and pathological RNAs have not been identified. Here, by ribosome profiling (Ribo-seq) we find specific dysregulation of endoplasmic reticulum (ER)-targeted mRNA translation in DIS3L2-deficient cells. Mechanistically, DMD functions in the quality control of the 7SL ncRNA component of the signal recognition particle (SRP) required for ER-targeted translation. Upon DIS3L2 loss, sustained 3'-end uridylation of aberrant 7SL RNA impacts ER-targeted translation and causes ER calcium leakage. Consequently, elevated intracellular calcium in DIS3L2-deficient cells activates calcium signaling response genes and perturbs ESC differentiation. Thus, DMD is required to safeguard ER-targeted mRNA translation, intracellular calcium homeostasis, and stem cell differentiation.

[1] Stem Cell Program, Division of Hematology/Oncology, Boston Children's Hospital, Boston, MA 02115, USA. [2] Department of Biological Chemistry and Molecular Pharmacology, Harvard Medical School, Boston, MA 02115, USA. [3] Manton Center for Orphan Disease Research, Boston, MA 02115, USA. [4] Section on Integrative Physiology and Metabolism, Joslin Diabetes Center, Harvard Medical School, Boston, MA 02215, USA. [5] Section on Islet Cell and Regenerative Biology, Joslin Diabetes Center, Harvard Medical School, Boston, MA 02115, USA. [6] Harvard Stem Cell Institute, Cambridge, MA 02138, USA. [7] Department of Pediatrics, Harvard Medical School, Boston, MA 02115, USA. [8] Harvard Initiative for RNA Medicine, Boston, MA 02115, USA. ✉email: rgregory@enders.tch.harvard.edu

While surveillance of protein-coding messenger RNAs (mRNAs) and mechanisms of nonsense-mediated decay (NMD) have been extensively investigated, little is known about quality control processes for non-coding RNAs (ncRNAs). DIS3L2-mediated decay (DMD) was recently identified as a surveillance pathway for certain ncRNAs, including ribosomal RNAs (rRNAs), transfer RNAs (tRNAs), small nuclear RNAs (snRNAs), microRNAs (miRNAs), and RNA component of mitochondrial RNA processing (RMRP)[1–13]. These aberrant ncRNAs are oligouridylated by the terminal uridyl transferases (TUTases) TUT4 (also known as ZCCHC11, and TENT3A) and TUT7 (ZCCHC6, or TENT3B), and subsequently degraded by the 3′–5′ exoribonuclease DIS3L2. Accordingly, DMD was implicated in a variety of cellular and physiological processes, including miRNA biogenesis and stability[1,11,14–16], rRNA maturation[9,13], mRNA degradation and apoptosis[17–19], cell proliferation[20–23], differentiation[14,15], and gametogenesis[24], as well as alternative splicing[22]. DIS3L2 binds to and degrades RNA species that are tagged by poly-uridine 3′ tails—a RNA modification that is catalyzed by the cytoplasmic TUTases[1,5,7,11,25]. Increasing evidence supports that this tag marks highly structured ncRNAs with imprecise 3′ ends for degradation, which could otherwise impair their proper folding and/or formation of multi-subunit ribonucleoprotein complexes[7–10,12]. One of the highly enriched ncRNAs in DIS3L2 immunoprecipitation (IP) experiments is the 7SL RNA component of the signal recognition particle (SRP) complex involved in endoplasmic reticulum (ER)-targeted mRNA translation[7,12]. While mutations in *DIS3L2* are associated with Perlman syndrome[20], the biological significance of impaired DMD for aberrant ncRNAs, including 7SL RNA, is obscure and pathological RNAs have not been identified[18,21,24,26]. Moreover, DMD contribution to mRNA translation has not been addressed so far.

Eukaryotic ER bound to translating ribosome machineries (also referred to as rough ER, or RER) is the main organelle responsible for coordinated biogenesis, folding, post-translational modification, and sorting of membrane-associated, secretory, and extracellular proteins[27–29]. Moreover, ER, with its unique architecture stretching from the nuclear envelope to the cell membrane[30] functions as a main intracellular storage reservoir for calcium ions ($Ca^{2+}$), responds to environmental cues and developmental signals and is involved in stress sensing in eukaryotic cells[31–34]. The biogenesis of several secreted growth factors and hormones, as well as membrane-localized signaling receptors, metabolites and ion channels, rely on ER-associated mRNA translation (reviewed in ref. [35]). Among other pathways, SRP-dependent recruitment of ribosome-bound mRNAs to the ER translocons is a major first step towards the final destination of the encoded proteins[36–41]. SRP itself is an evolutionarily conserved ribonucleoprotein complex comprising of the RNA polymerase III-encoded 7SL RNA as well as six protein subunits: SRPs 72, 68, 54, 19, 14, and 9 in eukaryotes. Notably, disruption of SRP complex results in dysregulation of ER-associated mRNA translation and secretory protein sorting[39], suggesting the significance of intact SRP complex for normal secretory and membrane proteins. ER-targeted mRNA translation starts with cytosolic ribosomes bound to respective mRNAs that stall upon translation of the signal peptide in the amino-terminus of the nascent polypeptide[40,42]. Signal peptide recognition and binding by SRP is essential for this stall and for recruitment of the mRNA to the ER membrane. Perturbation of SRP abrogates ER-targeted mRNA translation and results in inhibition of protein sorting or protein secretion[39,43,44], as well as increased calcium leakage from the ER translocon[45,46].

In this study, we reveal a key role for DMD-mediated quality control of 7SL RNA. In the absence of DIS3L2, the aberrant uridylated 7SL RNA inhibits the function of the SRP that leads to defective translation of secreted and transmembrane proteins at the ER and compromised ER-targeted calcium homeostasis. Consequently, embryonic stem cell (ESC) differentiation including that towards the renal lineage is perturbed, reminiscent of the renal abnormalities in Perlman syndrome patients[20].

## Results

**DIS3L2 is specifically required for ER-targeted mRNA translation.** We set out to simultaneously study mRNA expression and mRNA translation efficiency (TE) in *DIS3L2* knockout mouse ESCs (mESCs) using ribosome profiling (Ribo-seq)[47]. Consistent with previous reports[4,12], DIS3L2 loss did not affect global mRNA expression levels. Strikingly however, altered translation of many mRNAs was detected by changes in the abundance of ribosome-protected fragments (RPFs) (Fig. 1a, Supplementary Fig. 1a, and Supplementary Data 1). TEs of hundreds of mRNAs were significantly changed (at least 2-fold) in *DIS3L2* knockout cells compared to control ESCs (Fig. 1b). Gene ontology (GO) analysis of translationally downregulated mRNAs showed enrichment of membrane- and ER-localized transcripts, both of which utilize ER-associated mRNA translation for protein synthesis, whereas translationally upregulated transcripts were associated with the mitochondria (Fig. 1c). Since the connection between DIS3L2 function and mitochondrial physiology has been previously established[17,18], we chose to focus on the role of DIS3L2 in ER-associated protein synthesis. For the translationally downregulated mRNAs, we observed decreased RPFs throughout the coding regions (CDS) of the mRNAs, yet interestingly, we noticed a specific accumulation of RPFs at the 5′ end of the respective CDS (Fig. 1d), consistent with the observation that signal peptides are typically contained at the termini of respective proteins[48]. In contrast, RPFs of translationally upregulated mRNAs were uniformly increased throughout CDS (Supplementary Fig. 1b). Using three independent tools (Phobius[49], SignalP[50], TMHMM[51]), significantly more transmembrane or signal peptide-containing proteins were detected among the transcripts with downregulated translation in knockout cells (Fig. 1e). These results suggest that in *DIS3L2* knockout cells, translation of some of the ER-associated mRNAs is stalled at the sequences that encode signal peptide.

Metabolic labeling ($^{35}S$ methionine labeling) of de novo synthesized proteins revealed specifically reduced levels of secreted proteins in the culture media (but not in the cell lysates) of *DIS3L2* knockout cells (Fig. 2a, Supplementary Fig. 2a). Chemical inhibition of the sarco/ER $Ca^{2+}$ ATPase by thapsigargin treatment[52] suppressed de novo protein synthesis to a greater extent in *DIS3L2* knockout cells than in control cells (Fig. 2b), suggesting compromised ER-mediated calcium homeostasis upon DIS3L2 depletion. To directly assess ER-targeted mRNA translation, we used a specific luciferase reporter (GLuc) that is secreted from the cells and accumulates in the culture media[53]. Secreted luciferase (but not other luciferase reporters, Fig. 2f) was produced at a significantly lower level in DIS3L2-depleted mESCs (Fig. 2c). Re-expression of WT DIS3L2, but not the catalytic mutant protein, rescued the relative levels of secreted luciferase in the knockout cells (Fig. 2d). The specific requirement for DIS3L2 for expression of the secreted luciferase reporter was also detected in different human cell lines (Fig. 2e). Next, to investigate the signal peptide sequence requirement for defective ER-targeted translation in the absence of DIS3L2, we engineered dual luciferase reporters containing *Firefly* (as an internal control) and *Renilla* luciferases with or without ER-specific signal peptide (from *Insulin* mRNA) and an ER-retention signal, KDEL (http://signalpeptide.de). Assessing luciferase activities of these and the

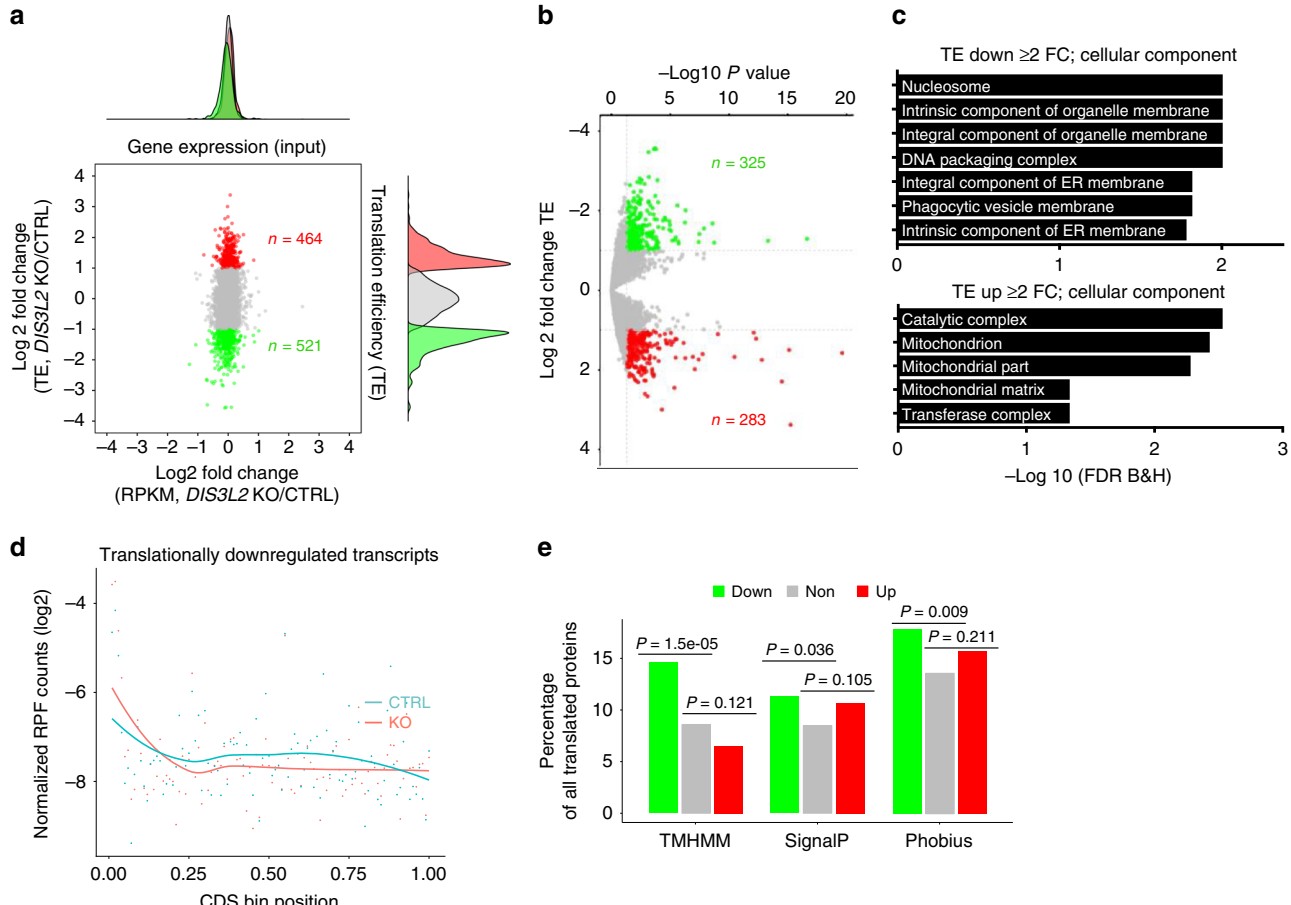

**Fig. 1 Dysregulated ER-targeted mRNA translation upon DIS3L2 loss. a** Scatter plot representing mRNA expression (input) and translation efficiency [ribosome-protected fragments (RPFs) abundance in *DIS3L2* knockout cells compared to heterozygote cells as control (CTRL)]. Translationally up- and downregulated genes (≥2-fold change) are marked in red and green, respectively. **b** Scatter plot representing significant perturbation of translation efficiency upon DIS3L2 loss. *P* values are from two-sided Wald tests (DESeq2) and *P* value adjustments were made for multiple comparisons. **c** Gene ontology analysis of translationally dysregulated mRNAs in the *DIS3L2* knockout mESCs. **d** Overall distribution of RPFs among translationally downregulated mRNAs. **e** Significant over-representation of transmembrane and/or signal sequences containing transcripts among translationally downregulated mRNAs (*P* values are from two-sided Fisher's exact tests).

parental reporters in the lysates and the supernatant samples obtained from control and *DIS3L2* knockout mESCs showed that defective protein translation in *DIS3L2* knockout cells depends on the presence of the signal peptide-coding sequence at the 5′ end of the luciferase reporter. Also, removal of the KDEL signal from the 3′ end of the reporter direct luciferase reporters more effectively to the supernatant and diminishes their detection in the lysates, thereby underscoring the robustness and sensitivity of our reporter systems (Fig. 2f). Notably, no tangible changes were observed in the mRNA expression of transfected luciferase reporters in *DIS3L2* knockout cells (Supplementary Fig. 2b–e). These results highlight the specific requirement of DIS3L2 for normal ER-targeted translation of a subset of mRNAs encoding transmembrane or secreted proteins.

**DIS3L2-mediated quality control of 7SL ncRNA.** DMD is involved in the quality control of several structured ncRNAs[4,7,10,12,21,24]. While DIS3L2 has been implicated in the surveillance of rRNAs, since we found no change in steady-state rRNAs levels, bulk translation as measured by polysome analysis[9] or metabolic labeling (Fig. 2a), or requirement for DIS3L2 for the translation of non-secreted luciferase reporters (Fig. 2f), we considered that rRNA is unlikely to account for the observed specific

defect in ER-targeted translation. We next explored whether defects in certain tRNAs might contribute to the defective translation detected in DIS3L2-deficient cells. Using our Ribo-seq data and published CLIP-seq results on DIS3L2-targeted tRNAs[12], we performed an analysis of codon usage frequencies in *DIS3L2* knockout mESCs. This showed no codon exclusion or preference associated with these potentially misprocessed tRNAs (Supplementary Figs. 3 and 4), which suggests that it is unlikely that impaired ER-targeted translation in *DIS3L2* knockout cells is caused by defective tRNA incorporation into ribosomes. Instead, due to a direct involvement of the 7SL RNA component of the SRP complex on ER-targeted translation[40,42], we sought to systematically determine whether defective ER translation is due to a role for DIS3L2 exoribonuclease in the quality control of 7SL RNA. Rapid amplification of complementary DNA (cDNA) ends from circularized RNAs (cRACE)[8] was used to precisely characterize the 3′ end of the 7SL RNA in DIS3L2-depleted cells. This showed extensive 3′-end uridylation in DIS3L2-bound transcripts (Fig. 3a), which are mostly truncated from the canonical 3′ end of 7SL RNAs (Fig. 3b). Logistic regression analysis of DIS3L2-bound 7SL RNAs determined a significantly positive correlation between the 7SL RNA 3′-end truncation and the probability of 3′-end uridylation, further underscoring the quality control function of DIS3L2, especially for truncated and thus aberrant 7SL RNAs (Fig. 3c).

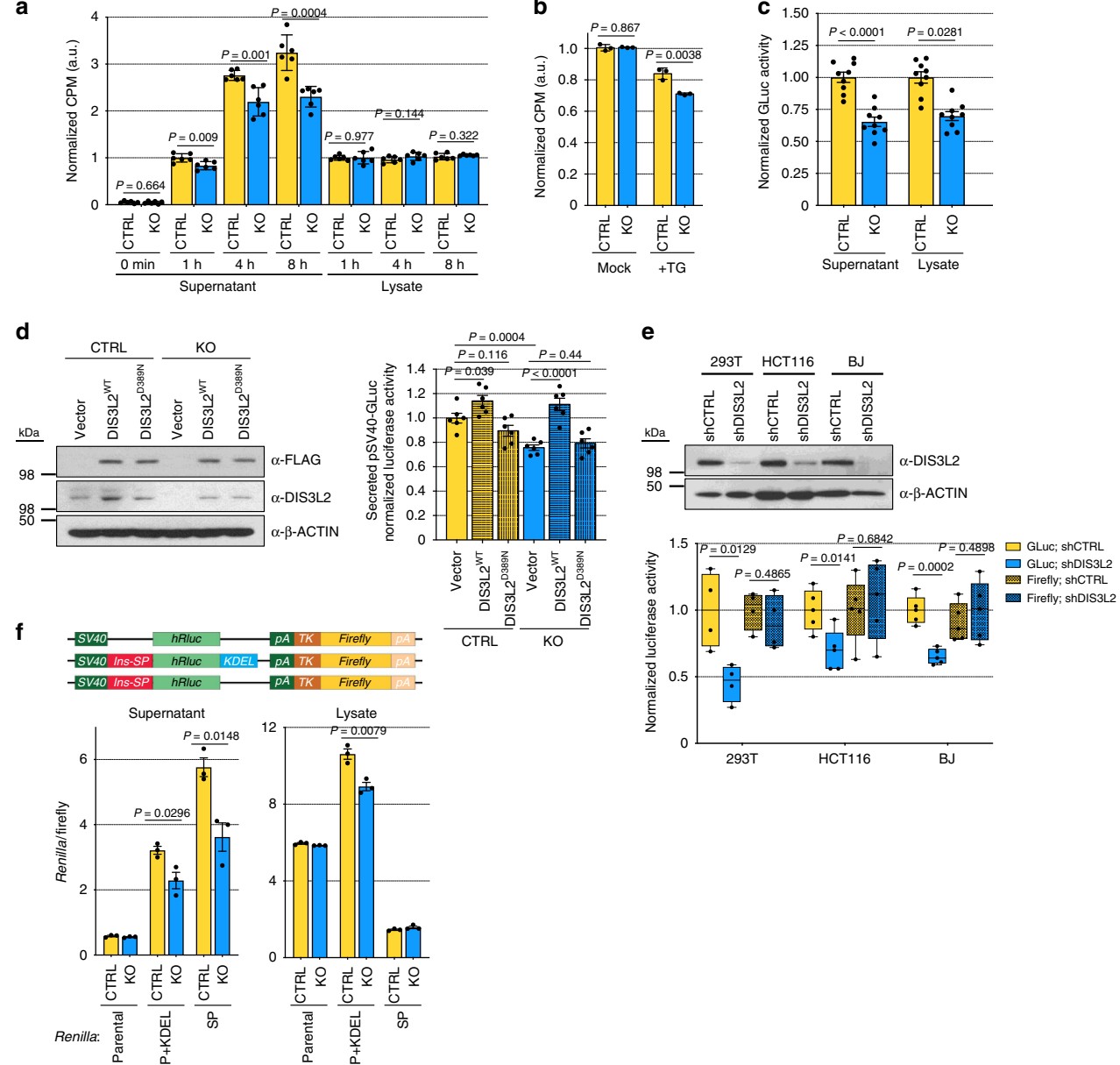

**Fig. 2 DIS3L2 is specifically required for ER-targeted mRNA translation. a** Quantification of de novo protein synthesis in cell supernatants or lysate samples using metabolic labeling. Bars representing mean ± SEM and *P* values (unpaired two-tailed Student's *t* test, 0.95 confidence intervals; *n* = 6 independent experiments) are shown. **b** Quantification of the de novo protein synthesis after TG treatment using metabolic labeling. Bars represent mean ± SEM and *P* values (unpaired two-tailed Student's *t* test, 0.95 confidence intervals; *n* = 3 independent experiments) are indicated. In **a**, **b**, CPM represents counts per minute. **c** Normalized GLuc luciferase activity in the supernatant or cell lysates. Bars representing mean ± SEM and *P* values (unpaired two-tailed Student's *t* test, 0.95 confidence intervals; *n* = 9 independent experiments) are indicated. **d** Left panel showing Western blot and right panel showing normalized activity of secreted GLuc reporter after over-expression of DIS3L2 proteins. Bars representing mean ± SEM and *P* values (unpaired two-tailed Student's *t* test, 0.95 confidence intervals; *n* = 6 independent experiments) are indicated. **e** Upper panel: Western blot analysis of DIS3L2 expression in different human cell lines stably depleted of DIS3L2; lower panel: normalized activities of ER (GLuc) and cytosolic (*Renilla*) luciferases after *DIS3L2* knockdown [*n* = 4 independent experiments (293T cells); *n* = 5 independent experiments (HCT116 cells) and *n* = 5 independent experiments for BJ cells]. Minimum, maximum, median, and boxes extending from the 25th to 75th percentiles are represented. *P* values (unpaired two-tailed Student's *t* test, 0.95 confidence intervals) are shown. Boxes mark minimum to maximum values. **f** Upper panel: Schematic representation of dual luciferase reporters psiCHECK-2 (parental), 5′-end signal peptide/3′-end KDEL-tagged (SP + KDEL), and only 5′-end signal peptide tagged (SP), respectively; lower panels: normalized activity of indicated luciferase reporter in the supernatants or lysate samples from control or *DIS3L2* knockout mESCs. Bars representing mean ± SEM and *P* values (unpaired two-tailed Student's *t* test, 0.95 confidence intervals; *n* = 3 independent experiments) are indicated. Source data are provided as a Source data file.

IP of FLAG-SRP68 protein and quantitative real-time PCR (qRT-PCR) showed uridylated 7SL RNA to be associated with the SRP (Fig. 3d), and IP of a ribosomal protein showed that uridylated 7SL RNA accumulates in ribosomes (Fig. 3e) in knockout mESCs. Northern blot analysis of total RNA samples showed accumulation of uridylated 7SL RNA in DIS3L2-deficient mESCs (Fig. 3f). Moreover, sucrose gradient centrifugation and qRT-PCR showed the presence of uridylated 7SL RNA in subpolysome and especially in polysomes in DIS3L2-deficient ESCs (Fig. 3g). Notably, defective ER-targeted translation in

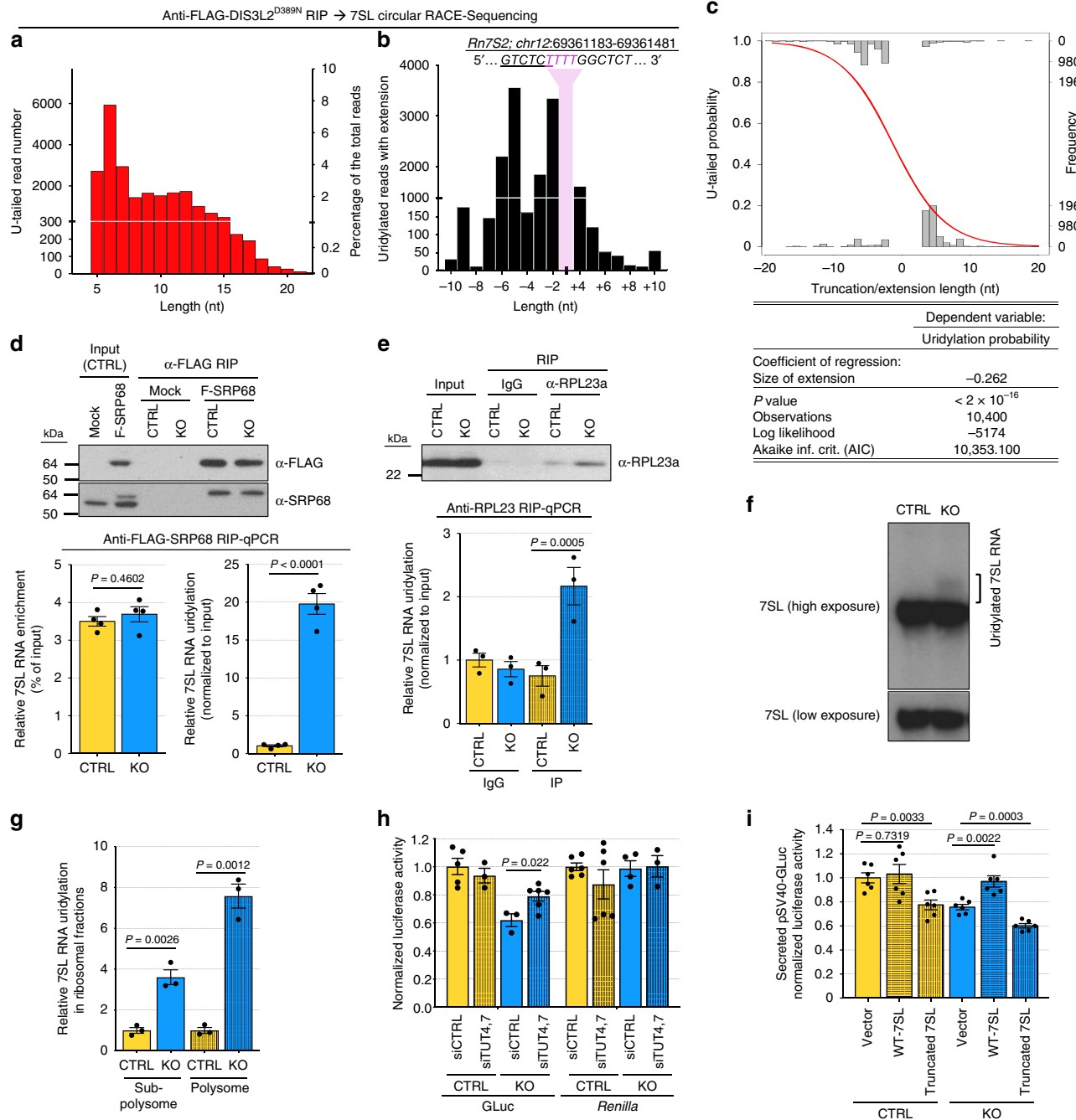

**Fig. 3 DIS3L2-targeted quality control of 7SL. a** cRACE analysis of DIS3L2-bound 7SL RNAs in knockout cells revealing extensive 3′-end uridylation. **b** cRACE analysis of DIS3L2-bound 7SL RNAs in respect to its canonical 3′ end (underlined). Note that most of the reads end before the canonical end. Nucleotides from mature 7SL RNA or genomic extensions that are transcribed into uridine are marked in magenta and excluded from the analysis. **c** Logistic regression analysis of U-tailed occurrence in respect to 7SL RNA truncation or genomic extension. **d** Co-precipitation of 7SL RNA with ectopically expressed FLAG-SRP68 (F-SRP68) protein analyzed by qRT-PCR. Upper panel showing Western blot. Lower left panel: Equal enrichment of 7SL RNA in FLAG-SRP68 precipitates from knockout and control samples; lower right panel: relative uridylation of 7SL RNA in corresponding samples. Bars represent mean ± SEM and P values (unpaired two-tailed Student's t test, 0.95 confidence intervals; n = 4 independent experiments). **e** Enrichment of uridylated 7SL RNA in RPL23a-IP samples in *DIS3L2* knockout samples. Upper panel showing Western blot and lower panel showing qRT-PCR. Bars representing mean ± SEM and P values (unpaired two-tailed Student's t test, 0.95 confidence intervals; n = 3 independent experiments) are indicated. **f** Northern blot analysis of 7SL RNA expression in total RNA samples from control and *DIS3L2* knockout cells. Slow migrating uridylated 7SL RNA species are marked. This experiment was repeated thrice independently with similar results. **g** qRT-PCR analysis showing a significant accumulation of uridylated 7SL RNA in the subpolysome and specially polysome fraction of *DIS3L2* knockout cells. Bars representing mean ± SEM and P values (unpaired two-tailed Student's t test, 0.95 confidence intervals; n = 3 independent experiments) are indicated. **h** Normalized activity of GLuc and Renilla (from TK vector) reporters in DIS3L2 knockout mESCs, and the effect of TUTases depletion on these two reporters. Bars represent mean ± SEM. P values (unpaired two-tailed Student's t test, 0.95 confidence intervals) are shown (n = at least three independent experiments). KO represents *DIS3L2* knockout mESCs and CTRL represents heterozygous control mESCs. **i** Normalized secreted luciferase activity following forced expression of full-length (WT) or truncated 7SL RNAs (n = 6 independent experiments). Bars representing mean ± SEM and P values (unpaired two-tailed Student's t test, 0.95 confidence intervals) are shown. Source data are provided as a Source data file.

DIS3L2 knockout cells was (at least partially) rescued by knockdown of the TUTases, TUT4/7, that mediate 7SL RNA uridylation (Fig. 3h, Supplementary Fig. 5a, b). Finally, forced expression of truncated 7SL RNA (Fig. 3i, Supplementary Fig. 5c–e) inhibited ER-targeted translation of the luciferase reporter mRNA, whereas over-expression of full-length [wild-type (WT)] 7SL RNA could rescue the defective ER-targeted luciferase translation in DIS3L2 knockout cells (Fig. 3i), which strongly supports the requirement of the DMD pathway in the surveillance of 7SL RNA and for ER-targeted mRNA translation.

**Impaired intracellular calcium storage and insulin release upon DIS3L2 loss.** Consistent with our Ribo-seq, metabolic labeling, and reporter assays showing defective ER-targeted translation, we found decreased calcium levels in the ER of DIS3L2 knockout cells with corresponding increased calcium ion ($Ca^{2+}$) concentration in the cytosol. This dysregulation of calcium homeostasis mainly resulted from the increased calcium leakage from the ER. Inhibition of translation elongation by anisomycin treatment was shown to lock the translocon in a $Ca^{2+}$-impermeable configuration[45,54]. Interestingly, anisomycin treatment alleviated ER calcium leakage in DIS3L2 knockout cells (Fig. 4a). Stimulation of ER calcium release using chemicals targeting different calcium channels further supports defective ER calcium mobilization in DIS3L2-depleted cells (Fig. 4b–e). While ER-targeted translation and ER function in intracellular calcium homeostasis are interconnected processes, our findings suggest that (i) DIS3L2 safeguards ER-targeted translation of transmembrane and secretory proteins and (ii) upon DIS3L2 depletion, the defective/delayed ER translation may cause ER calcium leakage an imbalanced intracellular localization of calcium.

Insulin secretion is a complex physiological process that relies on delicate regulation of calcium intracellular concentration. To test the effect of DIS3L2 loss on hormone secretion, we used Min6 cells as an in vitro model of pancreatic β-cells[55,56]. Min6 cells sense increased glucose levels and metabolize glucose, which in turn leads to increased ATP/ADP level that triggers closure of a potassium channel. This causes cell membrane depolarization, calcium ion influx, and finally insulin secretion. Upon transient DIS3L2 depletion, ER-targeted translation (Fig. 4f) and glucose-stimulated insulin secretion (Fig. 4g) were attenuated in Min6 cells. Moreover, upon bypassing glucose-sensing steps in Min6 cells by KCl administration, again DIS3L2-depleted cells failed to effectively secrete insulin (Fig. 4g). DIS3L2 knockdown in Min6 cells caused ER calcium depletion and increased cytosolic calcium (Fig. 4h), as well as increased calcium leakage from the ER (Fig. 4i) similar to DIS3L2 knockout mESCs, suggesting a common defect in the regulation of calcium homeostasis. Finally, in contrast to control cells, DIS3L2-depleted Min6 cells failed to increase cytosolic calcium level upon glucose stimulation (Fig. 4j). This highlights the requirement of DIS3L2 for endogenous protein secretion, at least partially through its role in ER-targeted translation to prevent calcium leakage from the ER.

**DIS3L2 is required for normal embryonic stem cell differentiation.** Regulation of calcium signaling is critical for proper stem cell differentiation and organ development, and its perturbation leads to malignancy and transformation[57–59]. Moreover, phenotypic analysis of DIS3L2 knockout animal models or Perlman syndrome[26] has suggested a differentiation failure in the renal lineage, although the molecular mechanism contributing to this phenotype is obscure. To elucidate the effect of DIS3L2 loss and calcium homeostasis in stem cells and during renal differentiation, we utilized spontaneous and renal lineage-directed

in vitro differentiation of DIS3L2-depleted mESC line (Supplementary Fig. 6a) carrying renal-specific Osr1-GFP reporter[60]. Similar to DIS3L2 knockout ESCs, stable DIS3L2 knockdown using specific short hairpin RNA (shRNA) led to TUTase-dependent downregulation of ER-targeted translation as measured by GLuc reporter (Supplementary Fig. 6b). DIS3L2 knockdown had no tangible effect on pluripotency of the mESCs, as was evident by normal proliferation, pluripotency marker expression, and their clonogenicity as compared to isogenic control knockdown ESCs (Supplementary Fig. 6c–e). However, DIS3L2 deficiency resulted in the formation of larger embryoid bodies (EBs) during spontaneous ESC differentiation (Fig. 5a, b). High-throughput sequencing of PolyA+ RNAs, hierarchical clustering, and principal component analysis (PCA) of RNA-sequencing data revealed extensive changes in mRNA expression of DIS3L2-deficient cells, especially at the later stages of EB differentiation (Fig. 5c–e, Supplementary Data 2 and 3). Moreover, consistent with defective ESC differentiation, dissociation and replating of cells from EBs at day 12 (d12) in the presence of leukemia inhibitory factor (LIF) and serum generated ESC colonies in DIS3L2-depleted cultures, but not in control knockdown ESCs (Fig. 5f). Interestingly, GO analysis of differentially expressed genes (DEGs) at d10 (Supplementary Fig. 6f, g) and d12 (Fig. 5g, h) of EB differentiation marked enrichment of specific molecular functions (e.g., calcium ion binding) and biological processes (e.g., extracellular matrix organization) (Fig. 5h, Supplementary Fig. 6g). Moreover, DEGs at d12 were associated with "abnormal appendicular skeleton morphology," "renal glomerular disease," "spontaneous abortion," and "wide anterior fontanel" (Fig. 5h), phenotypes that commonly occur in Perlman syndrome patients[20] and/or are related to calcium homeostasis. Together, these data suggest that while ESC pluripotency is unaffected, DIS3L2 deficiency causes late differentiation defects that are associated with calcium homeostasis, as well as bone and renal development, recapitulating some of the developmental phenotypes that characterize Perlman syndrome patients.

**DIS3L2 loss perturbs calcium signaling pathways and renal differentiation.** Time-course analysis of gene expression during EB differentiation showed increased expression of $Ca^{2+}$ signaling target/sensor genes in DIS3L2 knockdown cells. These include polycystic kidney disease-associated proteins (Pkd1, Pkd2), calcium/calmodulin-dependent protein kinases (Camk genes), and Regucalcin, but not randomly selected genes (Fig. 6a, Supplementary Data 3). This elevated expression of calcium signaling targets or sensors suggests the cellular response to perturbed calcium homeostasis in DIS3L2-depleted cells (Fig. 4). Moreover, in shDIS3L2 cells, the expression of renal-associated transcription factors was dysregulated during ESC differentiation (Fig. 6b, Supplementary Data 2 and 3). This prompted us to specifically investigate the renal differentiation propensity of DIS3L2-deficient cells using renal-directed differentiation[60] (Fig. 6c). DIS3L2 knockdown caused larger renal differentiation aggregates and increased proportion of renal progenitor cells as evidenced by the increased number of Osr1-GFP-expressing cells (Fig. 6d, e, Supplementary Fig. 7), and also the increased expression of renal-specific markers of the metanephric mesenchyme progenitor stage (Fig. 6f, g). These observations are in line with organ—and especially kidney—overgrowth and developmental defects observed in Perlman syndrome patients. Finally, to address the casual and pathological effects of dysregulated 7SL RNA decay in DIS3L2-depleted cells, on renal differentiation, we stably over-expressed truncated 7SL RNA in control and DIS3L2 knockout ESCs and induced in vitro renal differentiation. qRT-PCR

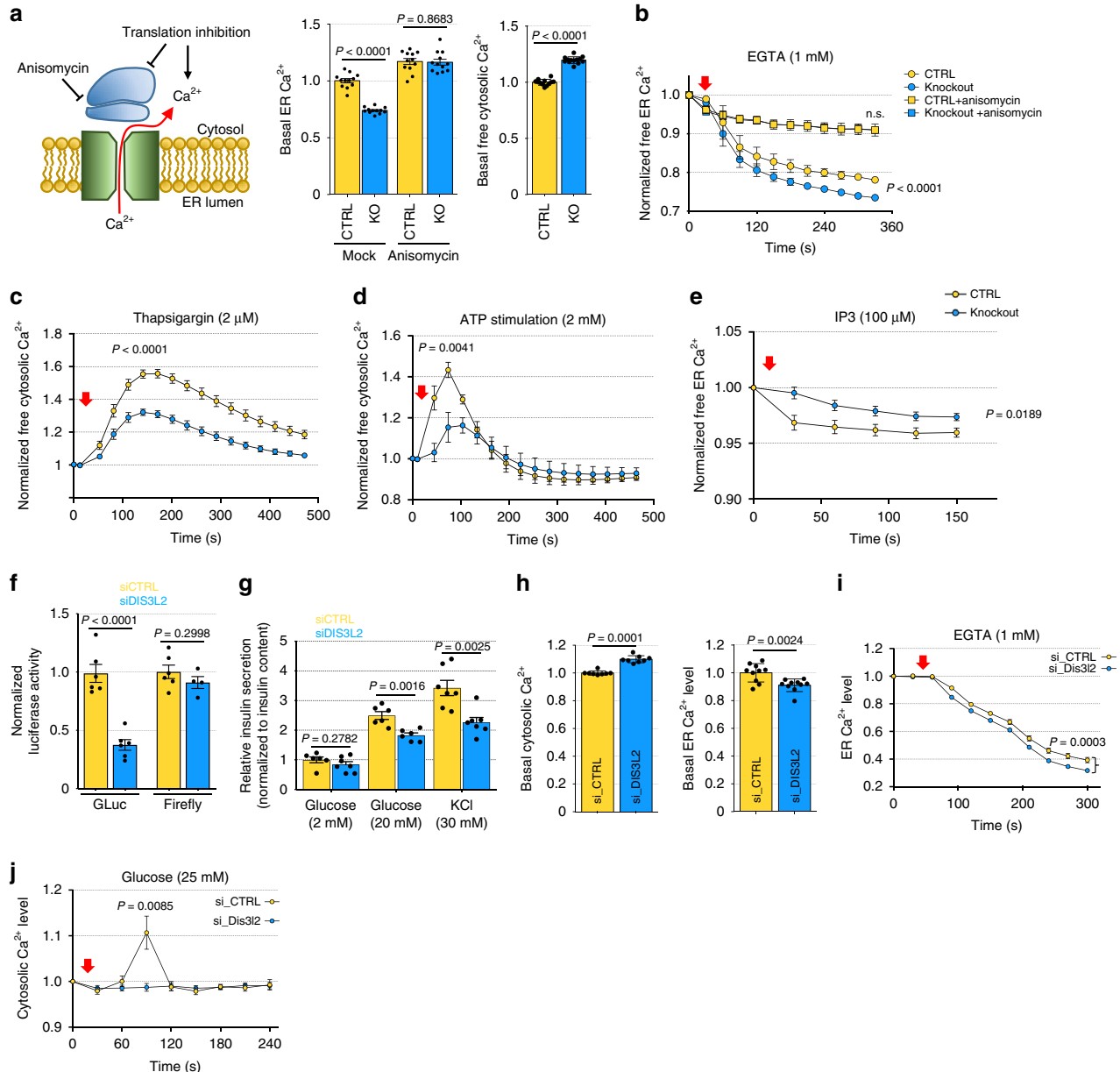

**Fig. 4 Impaired intracellular calcium storage upon DIS3L2 loss. a** Left panel: Schematic representation of increased ER calcium leakage through ER membrane-localized translocon (shown in green) upon translation inhibition (perhaps by DIS3L2 loss). Basal free ER (middle) and cytosolic (right) $Ca^{2+}$ levels are shown in the middle and right panels, respectively ($n = 12$ independent experiments). Bars representing mean ± SEM and $P$ values (unpaired two-tailed Student's $t$ test, 0.95 confidence intervals) are indicated. **b**–**e** ER calcium leakage in mESCs pretreated with or without anisomycin ($n \geq 8$). In **b**, note the overlapping values for control and knockout cells treated with anisomycin. Normalized free cytosolic $Ca^{2+}$ measured after treatment with thapsigargin (**c**) or ATP (**d**). **e** ER calcium release in mESCs upon IP3 stimulation (in **b**–**e**, $n = 10$ independent experiments). Red arrows indicate the time of chemical treatments. **f** Normalized luciferase activity of indicated reporters in Min6 cells ($n = $ at least four independent experiments). **g** Relative insulin secretion levels in Min6 cells upon glucose or potassium stimulations ($n = 6$ independent experiments). **h** Basal free ER (left) and cytosolic (right) $Ca^{2+}$ levels in Min6 cells transiently transfected with indicated siRNAs ($n = 8$ independent experiments). **i** ER calcium leakage in DIS3L2-depleted Min6 cells ($n = 10$ independent experiments); red arrow indicates the time of EGTA treatment. **j** Cytosolic calcium levels upon high glucose stimulation of siDIS3L2-transfected Min6 cells ($n = 10$ independent experiments); red arrow indicates the time of glucose stimulation. Bars represent mean ± SEM. $P$ values (unpaired two-tailed Student's $t$ test, 0.95 confidence intervals). Source data are provided as a Source data file.

analysis of renal-specific markers at the end of differentiation showed that (1) similar to *DIS3L2* knockdown cells, *DIS3L2* knockout cells also express higher levels of renal progenitor markers compared to control cells; and more interestingly, (2) over-expression of truncated and therefore aberrant 7SL RNA worsened this defect in *DIS3L2* knockout cells as is marked by elevated expression of Six2 and IGF2, as well as calcium-sensing protein Camk1g (Fig. 6h). This further highlights the essential

function of DIS3L2 in the regulation of renal differentiation and also connects the pathological role of truncated 7SL RNA to the defective differentiation of DIS3L2-deficient cells. Thus, while DIS3L2-depleted undifferentiated ESCs behave normally, they manifest differentiation defects associated with kidney development/function and show overgrowth, especially during renal lineage differentiation. This is also consistent with organomegaly and kidney hypoplasia phenotypes seen in Perlman syndrome

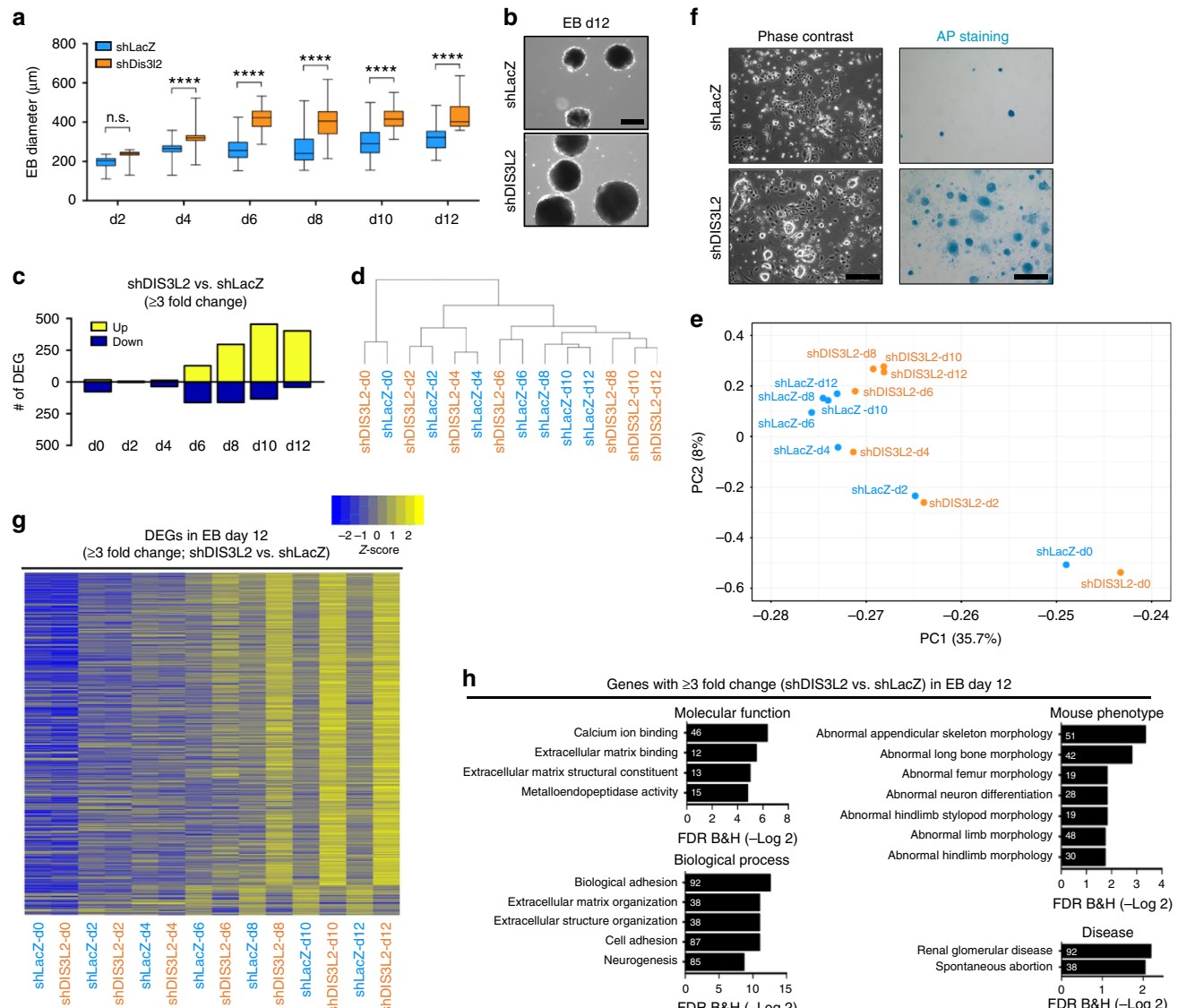

**Fig. 5 Impaired in vitro differentiation upon DIS3L2 loss. a** The size of EBs during in vitro spontaneous differentiation. Minimum, maximum, median, and boxes extending from the 25th to 75th percentiles are shown (n = at least 30 EBs measured). **b** Representative images of EBs at d12. This experiment was repeated twice independently with similar results. Scale bar = 200 μm. **c** The number of differentially expressed genes (DEGs) during spontaneous differentiation of EBs. Hierarchical clustering (**d**) and PCA analysis (**e**) of mRNA sequencing data are shown. **f** Phase contrast (left panels) and AP staining (right panels) of d12 differentiated cells replated for 5 days in the presence of LIF and serum to support ESC colonization. This experiment was repeated twice independently with similar results. Scale bar = 200 μm. **g** Heatmap representation of differentially expressed genes at d12 during the entire course of differentiation. **h** Gene ontology analysis of DEGs from **g** at d12 of differentiation; numbers inside the bars indicate the number of DEGs associated with each biological terms.

patients, and suggests that this is likely due to cell-autonomous abnormalities that can be recapitulated ex vivo.

## Discussion
In this study, we used cellular models to examine the physiological requirement of DMD and to illuminate the underlying molecular mechanism. We conclude that DIS3L2 functions to eliminate aberrant 7SL ncRNA subunit of the SRP particle, and thereby safeguards the ER-targeted mRNA translation of membrane-targeted and/or secretory proteins. Moreover, in the absence of DIS3L2, cellular calcium homeostasis and ER function is impaired. Besides the housekeeping functions, ER-associated mRNA translation and ER calcium homeostasis are particularly critical for the proper function of organs that are specialized in

physiological communication with other organs or with the environment. These include pancreas (secreting endocrine hormones and exocrine enzymes)[61], nervous system (functionally dependent on ion channels and synaptic vesicles)[62,63], muscles (relying on calcium release from ER to contract)[64,65], and kidney (harboring several important ion channels, involved in metabolite re-absorption)[66]. Notably, these are among the most affected organs in Perlman syndrome patients with defective DMD. Thus, it would be of interest to study ER dysfunction in patient samples and *DIS3L2* animal models of Perlman syndrome to determine the physiological relevance of defective DMD for 7SL RNA in tissue function and homeostasis[21,26].

We demonstrate that DIS3L2 functions in the quality control of truncated 7SL RNA. Several other DIS3L2 targets are also directly involved in the mRNA translation, including tRNAs[2,10,12] and

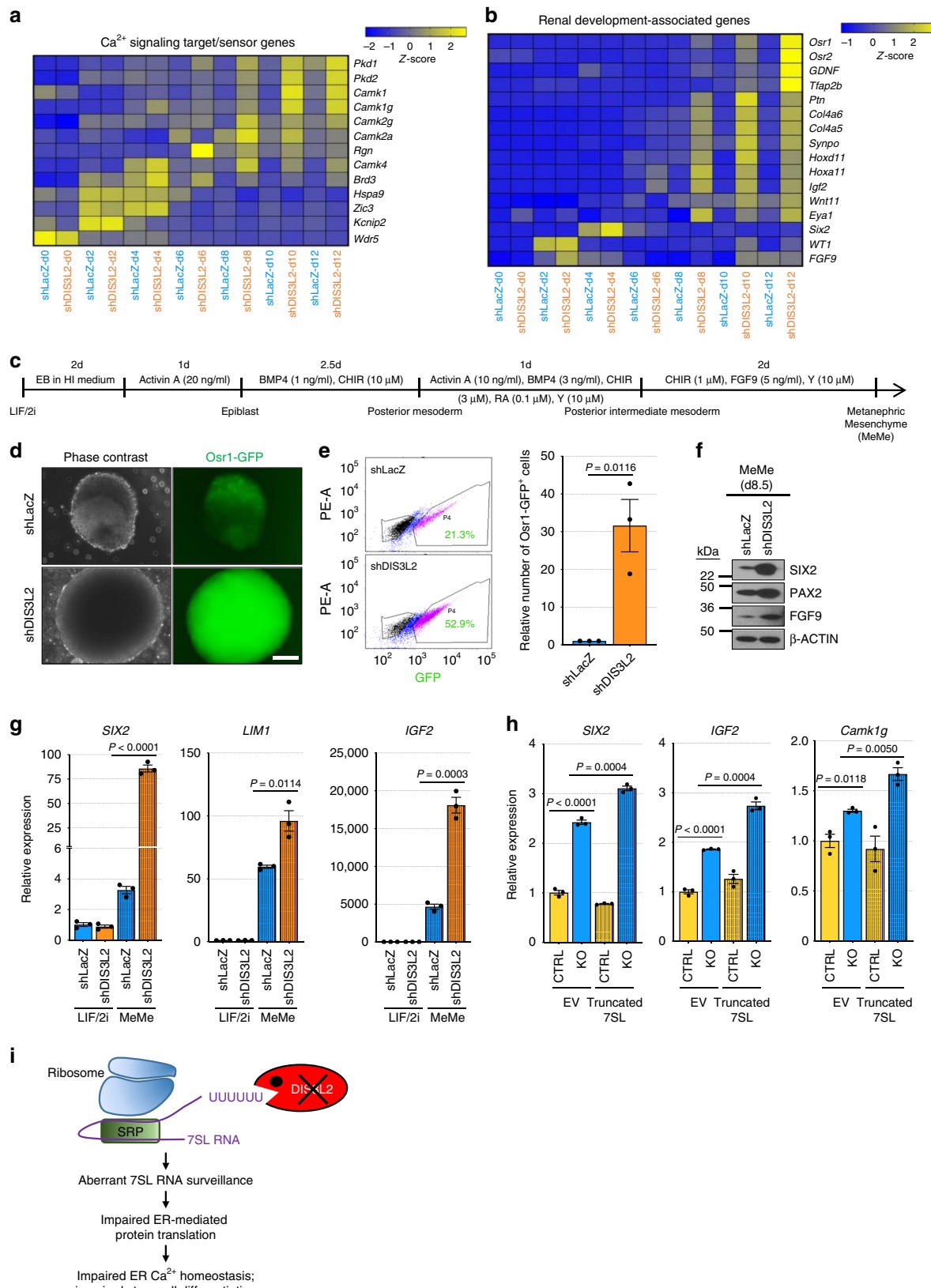

rRNAs[9,13]. However, our ribosome footprinting results point to a specific requirement for DIS3L2 in ER-targeted translation via elimination of aberrant 7SL RNAs with imprecise 3′ ends. Re-introducing WT (full-length) 7SL RNA into *DIS3L2* knockout cells partially rescued ER-targeted translation. Moreover, we did not observe any codon usage differences upon DIS3L2 loss.

Accordingly, we conclude that, at least in mESCs, defective ER translation is not caused by aberrant rRNAs or tRNAs. Instead, in the absence of DIS3L2, pathological truncated 7SL RNA incorporates into SRP particles, associates with ribosomes, and adversely affects homeostatic translation of several proteins, including membrane-targeted and secretory proteins. Cryo-electron

**Fig. 6 Altered calcium-sensing and renal differentiation propensity upon DIS3L2 loss. a** Heatmap representation of differentially expressed genes associated with calcium signaling or calcium sensing. **b** Heatmap representation of differentially expressed genes associated with renal differentiation. **c** The protocol used during in vitro differentiation of mESCs to renal progenitors (metanephric mesenchyme). **d** Phase-contrast and fluorescent images of Osr1-GFP^positive renal progenitors at d8.5 of renal differentiation. Note the larger size of differentiation aggregate as well as increased intensity of Osr1-GFP signal in shDIS3L2 culture. This experiment was repeated thrice independently with similar results. Scale bar = 100 μm. **e** Left panels: Flow cytometry analysis of Osr1-GFP^positive cells in samples from **d**; right panel, relative number of Osr1-GFP^positive cells at d8.5 of renal differentiation. Bars representing mean ± SEM and P values (unpaired two-tailed Student's t test, 0.95 confidence intervals; n = 3 independent experiments) are shown. **f** Western blotting analysis of samples from **d**; Met. Mes., metanephric mesenchyme stage, that is, 8.5 days after differentiation. Representative images of two independent experiments with similar results are provided. **g** qRT-PCR analysis of renal-specific markers for indicated samples at day 8.5 of differentiation, corresponding to the metanephric mesenchyme (MeMe) stage. Bars represent mean ± SEM and P values (unpaired two-tailed Student's t test, 0.95 confidence intervals; n = 3 independent experiments). **h** qRT-PCR analysis of indicated mRNAs samples at day 8.5 of differentiation, corresponding to the metanephric mesenchyme (MeMe) stage. Bars represent mean ± SEM and P values (unpaired two-tailed Student's t test, 0.95 confidence intervals; n = 3 independent experiments). Control and *DIS3L2* knockout cells were transduced with pLKO.1 empty vector (EV) or a vector stably expressing truncated 7SL RNA under its endogenous promoter. **i** A model describing the biological function of DIS3L2 in safeguarding ER-targeted translation, calcium homeostasis, and stem cell differentiation. Source data are provided as a Source data file.

microscopy and biochemical approaches have revealed that, at the nascent chain exit site, SRP contacts RPL23 in the large ribosomal subunit to bind the signal sequence[40,42]. Moreover, SRP perturbation leads to ER-translation defects and mistargeting ER proteins to the mitochondria[39]. The translation stall at the 5′ end of respective mRNAs that we observe in *DIS3L2* knockout cells likely explains the increased ER calcium leakage and consequently elevated cytosolic calcium ion levels.

The quality control function of DIS3L2 on 7SL RNA suggests that only a small fraction of the 7SL RNA pool in *DIS3L2* knockout cells is truncated and uridylated. The remaining question is how this small portion of 7SL RNA species, although aberrant, negatively impact ER-targeted translation? Also, it needs to be further investigated why only a proportion of mRNAs predicted to be translated through ER is affected by DIS3L2 loss. Truncated and/or uridylated 7SL RNA species may have altered affinity to protein subunits comprising the SRP complex. This, even at a low concentration, may cause a dominant-negative effect on SRP function in recruitment of mRNA/translating ribosome machinery to ER, on SRP complex turnover, or on the robustness and specificity of mRNA substrate recognition by SRP complex[36]. It is worth noting that not all the ER-translated mRNAs are SRP clients. In fact, a considerable amount of ER-targeted translation is SRP independent[39], possibly explaining why in DIS3L2-deficient cells only a subset of ER-associated mRNAs are affected. Furthermore, elevated translation of a subset of mitochondrial transcripts upon DIS3L2 loss as well as the pathological roles of other DIS3L2 targets, including rRNAs and miRNAs, should be investigated in future.

The physiological connection between intracellular calcium homeostasis and cell proliferation, as well as differentiation, has been extensively studied (reviewed in ref. [67]). For example, in lateral mesoderm retinoic acid induces the translocation of calcium channel TRPP2 from ER to the cell membrane, which in turn potentiates calcium influx to the cytosol, a signaling pathway that leads to the transcriptional activation of renal progenitor factor Pax8[68–70]. Accordingly, temporal and spatial regulation of calcium signaling is in particular critical for renal lineage development, in which $Ca^{2+}$ stimulates lateral mesoderm to differentiate into renal progenitors[69,70]. The altered differentiation propensity of DIS3L2-depleted cells toward the renal lineage could therefore be explained by their elevated intracellular $Ca^{2+}$ levels as well as their impaired intracellular storage mechanisms. Moreover, regulation of intracellular calcium homeostasis has been recently shown to be critical for exit from pluripotency and differentiation[59]. While LIF/2i-adapted DIS3L2-deficient ESCs remain undifferentiated, the greater competence of these cells to respond to differential stimuli after an in vitro, and perhaps

in vivo, exit from pluripotency results in enhanced renal lineage differentiation program and eventually leads to kidney overgrowth, similar to that in the Perlman syndrome patients. This highlights the relevance of DIS3L2 loss-induced defects we observe and the significance of stem cells as a tool for in vitro disease modeling. Finally, our findings raise the possibility that defective SRP complex function caused by DIS3L2 deficiency might be responsible for the severe developmental phenotypes associated with Perlman syndrome[20], as well as the perinatal lethality reported in *DIS3L2* knockout mice[26]. We therefore propose that compromised DMD leads to a "SRPopathy" marked by defective ER-targeted mRNA translation (Fig. 6i). Further investigations are required to determine the extent by which the proposed SRPopathy may contribute to the phenotypes observed in Perlman syndrome patients.

## Methods

**Cell culture.** TC1 mESCs were cultured without feeder on 0.2% gelatin (Sigma) as previously described[71]. *DIS3L2* knockout ES cells were generated previously using CRISPR/Cas9 gene editing[7]. ESCs were cultured in LIF/serum medium [containing Dulbecco's modified Eagle's medium (DMEM) (Gibco), 1000 U/ml mouse LIF (Gemini), 15% Stem Cell Qualified fetal bovine serum (FBS) (Gemini), 2 mM HEPES (Gibco), 1 mM sodium pyruvate (Gibco), 1× non-essential amino acid (NEAA) (Gibco), 2 mM L-glutamine (Gibco), 50 μM 2-mercaptoethanol (Thermo Fisher), and 1% penicillin–streptomycin (Gibco)] on 0.2% gelatin-coated dishes without feeders. Min6 cells were cultured in DMEM high glucose (Gibco) supplemented with 15% FBS (Gemini), 0.5% β-mercaptoethanol (99%) (ACROS), and 1% penicillin–streptomycin (Gibco). All the other human cancer cell lines were cultured in DMEM (Gibco) supplemented with 10% FBS (Gemini) and 1% penicillin–streptomycin (Gibco). Mouse embryonic fibroblast cultures were prepared from embryonic tissues at E12.5 according to standard protocol as described previously[71]. To prepare ex vivo cultures of E18.5 embryonic kidney and liver, tissues were isolated under aseptic condition, rinsed twice in phosphate-buffered saline (PBS), dissected into ~1–5 mm³ pieces, and incubated for 10–15 min with collagenase type IV (Stem Cell Technologies; 1 mg/ml) with occasional pipetting. Later, the cells were centrifuged and washed twice with cold PBS to remove the trace of enzymatic digestion. Finally, single cells were cultured on 6-well dishes and fed by ex vivo culture medium [1:1 DMEM/F12 (Gibco) and Neurobasal medium (Gibco) supplemented with 1× N2 and B27 supplements (Gibco), 10% FBS (Gemini), 1 mM sodium pyruvate (Gibco), 1× NEAA (Gibco), 2 mM L-glutamine (Gibco), 50 μM 2-mercaptoethanol (Thermo Fisher), and 1% penicillin–streptomycin (Gibco)]. Alkaline phosphatase staining of the mESCs was performed as previously described[72].

**Transfection and ER-reporter assays.** For transient knockdown experiments, following ON-TARGET plus siRNAs (small interfering RNAs) (all Dharmacon) were used: control siRNA pool (D-001810-10), siTUT7 (L-056770-01), and siTUT4 pool (L-065226-00). ESCs were reverse transfected using siRNAs and Lipofecta-mine RNAiMax (Invitrogen) complexes prepared in Opti-MEM (Gibco) for 48–72 h. For ER-reporter translation assays, 1 μg GLuc reporter plasmid was transfected into $10^6$ ESCs in 6-well dishes. At 12 h after the transfection, cells were washed twice, and supplemented with 1 ml fresh ESC medium. After 2–6 h, the conditioned media were collected and cells were washed twice with PBS and then lyzed in 1× Passive Lysis Buffer (Promega). Some cultures were alternatively treated with

2 μM thapsigargin or equal volume dimethyl sulfoxide (DMSO) (mock) and incubated for 2 h before analysis. Relative luciferase activities in both supernatants and lysates were analyzed using BioLux Gaussia Kit. As negative controls, psiCHECK-2 (Promega) or pRL-TK (Addgene) vectors were transfected and analyzed similarly to GLuc reporter. Values were normalized to total protein and/ or DNA content of the cultures. DIS3L2-stable knockdown ESCs (shDIS3L2 line) were generated previously[1]. Full-length (WT) and truncated 7SL RNAs were amplified from mouse genomic DNA and cloned into pGEM-T easy (Promega) or U6 promoter-less pLKO.1 (Sigma) vectors (see Supplementary Table 1 for the oligos used for cloning) and sequenced. For rescue experiment, 1 μg of these plasmids were transfected in mESCs for 48 h, after which the cells were re-transfected with pSV40-GLuc ER-reporter plasmid. For the negative control transfection, empty pGEM-T easy vector was ligated, and transformed into competent cells, from which maxiprep (Qiagen) preparation of plasmid was performed. Western blotting analysis was performed as previously described[9]. Following antibodies were used: anti-FLAG (Sigma; 1:10,000 dilution); anti-DIS3L2 (Novusbio; 1:1000 dilution); anti-β-ACTIN (Abcam; 1:5000 dilution); anti-SRP68 (Proteintech Group; 1:1000 dilution); anti-RPL23a (Proteintech Group; 1:1000 dilution); anti-SIX2 (Abcam; 1:500 dilution); anti-PAX2 (a kind gift from Dr. Kreidberg; 1:1000 dilution); anti-FGF9 (Abcam; 1:1000 dilution).

**RNA extraction and qRT-PCR.** Cells were washed twice with PBS, lyzed in Trizol (Ambion), and RNAs were chloroform–isopropanol extracted and washed twice with 70% ethanol. Two micrograms of RNA was treated with RQ1 DNase for 30 min at 37 °C. Using random hexamers (to analyze relative expression) or oligo-dA$_{12}$ oligonucleotides (to measure relative uridylation), cDNAs were made with SuperScript III reverse transcriptase (Invitrogen) and RNaseOUT (Invitrogen). List of primers and oligos are provided in Supplementary Table 1. All the qRT-PCR experiments were normalized to β-Actin levels in the respective cDNA samples. Northern blotting of 7SL RNA was performed as previously described[7].

**ESC differentiation.** ESCs were maintained in LIF/2i culture condition before differentiation as previously described[71]. For spontaneous differentiation, EBs were formed in hanging drops using 500 ESCs for 2 days, and then transformed to polyHEMA-coated (Sigma) (20 mg/ml in 75% ethanol) dishes and maintained in suspension for indicated timepoints before analysis. Differentiation was performed using ES medium depleted of LIF and 2i, and instead, supplemented with 15% FBS (Gemini). For renal differentiation, a previously described method[60] was used with slight modification: at the first step of differentiation (epiblast differentiation), 1000 ESCs were induced for 2 days to form aggregates in hanging drops containing HI medium [75% Ham's F12 medium and 25% Iscove's modified Dulbecco's medium (both from Gibco)] and then aggregates were transferred to polyHEMA-coated dishes and further induced for renal differentiation as described before[60]. At 8.5 days after differentiation (at metanephric mesenchyme stage), cell aggregates were harvested for qRT-PCR or western blot analysis. Alternatively, cell aggregates were dissociated with type 1 collagenase 1.5 mg/ml (Worthington Biochemical) for 10 min at room temperature with gentle shaking. Dissociated cells were centrifuged, washed with PBS (Invitrogen), and resuspended in 500 μl PBS and analyzed by flow cytometry (BD FACSAria). Undifferentiated mESCs (not expressing GFP) were used as the negative control.

**Metabolic labeling.** Equal numbers of control and DIS3L2 knockout mESCs were plated overnight and then washed with PBS and incubated in methionine- and cysteine-free DMEM (Gibco) medium for 2 h. Some cultures were alternatively treated with 2 μM thapsigargin or equal volume of DMSO (mock). Cells were then incubated for 1 h after supplementation with [$^{35}$S]-methionine ([$^{35}$S]-Met; 100 mCi/ml; PerkinElmer), after which they were washed with PBS to eliminate free radiolabeled amino acids. Total protein lysates were collected and the concentration of the proteins was measured by using Bradford assay. To measure radiolabeled secreted proteins, cells were starved in methionine- and cysteine-free ESC medium (Gibco) for 1 h, incubated for 1 h with [$^{35}$S]-Met containing ESC medium, washed twice with PBS, and then supplemented with fresh ESC medium (without additional [$^{35}$S]-Met) and the medium was collected immediately (0 min, to measure background, i.e., free amino acids) or after 1, 4, or 8 h (Supplementary Fig. 2a). For the quantitation of [$^{35}$S]-Met-labeled proteins in the lysates or in the supernatant medium, [$^{35}$S]-Met-labeled proteins were subjected to liquid scintillation analysis.

**RNA immunoprecipitation.** mESCs were transfected with FLAG-WT DIS3L2, FLAG-mutant DIS3L2[1], or empty pFLAG-CMV2 (as mock) vectors. At 48 h after transfection, mESCs were ultraviolet crosslinked, lyzed, and then RNA IP was performed using anti-FLAG M2 Affinity Gel beads (Sigma) as previously described[7], and co-precipitated RNAs were isolated, purified, and analyzed by qRT-PCR. Thirty micrograms of FLAG-SRP68 vector (OriGene Technologies, MR204949) was transfected into 10$^7$ ESCs in 15-cm dishes of DIS3L2 heterozygote (control) or knockout ESCs in triplicates using Lipofectamine 2000 (Invitrogen) overnight before changing the medium. At 48 h after transfection, cells were harvested without crosslinking and FLAG-SRP68 was precipitated using anti-FLAG M2 Affinity Gel beads (Sigma). Rabbit anti-RPL23a (Proteintech Group; 10 μg

antibody per IP) or normal rabbit IgG (Cell signaling) and protein A agarose beads (Roche) were used for IP of ribosomes in control and DIS3L2 knockout ESCs. Co-precipitated RNAs were isolated, purified, and analyzed by qRT-PCR.

**cRACE.** Input and FLAG-mutant DIS3L2 IP RNA samples were circularized with 1 μl T4 RNA ligase I, 10 mM ATP, and 10% PEG 8000 in 1× T4 RNA ligase buffer for 2 h at 37 °C and then the ligase was inactivated, as previously described[8]. After DNase I treatment, circularized RNAs were reverse transcribed with 7SL RNA-specific reverse primer (Supplementary Table 1) and SuperScript III. cDNAs were amplified by divergent internal primers (Supplementary Table 1) and AccuPrime GC-rich DNA Polymerase (Invitrogen) to generate chimeric PCR products corresponding to 5′ and 3′ ends of 7SL RNA transcripts. PCR products were size selected on 2% agarose gel, purified, and used in library preparation for MiSeq analysis using TruSeq Stranded mRNA Sample Preparation Kits (Illumina). To evaluate uridylation in RIP sample, sequencing reads containing ≥5 consecutive uridines (UUUUU) were considered as uridylated reads, which distinguishes them from reads corresponding to genomically extended 7SL RNA species with four uridines (Fig. 3a, b).

**mRNA sequencing.** PolyA$^+$ mRNAs were isolated at indicated timepoints from differentiated EBs and cDNA libraries were prepared using Illumina TruSeq® Standard Total RNA Sample Preparation Guide and sequenced through Illumina NextSeq 500 Sequencing pipeline. Sequencing reads were aligned to a reference genome built by Subread v 1.6.3[73] using ENSEMBLE mouse genome v88 (GRCm38), and then the transcripts were quantified by featureCounts v 1.6.3[73]. The normalization was performed using edgeR[74] employing counts per million normalization with R version 3.6 and RStudio v 1.2.1335. GraphPad Prism software was used for data presentation.

**Ribosome profiling.** Ribosome profiling was performed in two biological replicates of ESC cultures using TruSeq® Ribo Profile (Mammalian) Kit from Illumina. After adapter trimming and quality filtering, the sequences were further filtered to exclude rRNA and tRNA reads by aligning to mouse rRNA and tRNA sequence, which were retrieved from Ensembl (Release 91) (https://useast.ensembl.org/Mus_musculus/Info/Index) and GtRNAdb databases[75], respectively. The cleaned RPFs were aligned to the mouse canonical known gene models (UCSC, mm10) using Bowtie with a maximum one mismatch allowed[76]. For codon occupancy analysis, the A site were inferred using an offset of 15 nucleotides (nt) from 5′ end of 28–31 nt fragments, which were uniquely mapped and translated in the zero frame of CDS[77]. The codon occupancy of an A site was further normalized by its basal occupancy, which is the average codon occupancy among +1, +2, and +3 downstream of the A site[77]. The TE was calculated by dividing RPF abundancy on CDS by its mRNA abundancy. We used 2-fold change of TE as the threshold to define the differential translation genes[78,79]. For the metagene analysis, we divided each CDS region into 50 equal bins and counted the RPF occupancy in each bin, which were then normalized by total number of RPFs of respective gene.

**Measurement of cytosolic calcium.** Cytosolic calcium measurement was performed by using FLUOFORTE Calcium Assay Kit (Enzo). Cells in the 96-well plates were stained with 100 μl Hanks' balanced salt solution (HBSS) buffer containing FLUOFORTE-AM and 4′,6-diamidino-2-phenylindole (DAPI) at room temperature for 1 h. After washing and changing fresh HBSS buffer, basal cytosolic calcium level was determined by measuring the fluorescence emitted at 525 nm after the cells were excited at 490 nm and normalized by DAPI signal. Later, cells were stimulated by different chemicals (1 mM EGTA, 2 mM ATP, 2 μM thapsigargin, 100 μM IP3) and monitored the change of the fluorescence every 20–30 s for 5–8 min.

**Measurement of ER calcium.** ER calcium measurement was performed using Mag-Fluo4 acetoxymethyl ester (Mag-Fluo4-AM, Molecular Probes). Cells in the 96-well plate were stained with 5 μM of Mag-Fluo4-AM and DAPI at 37 °C for 40 min. After washing with the buffer (125 mM KCl, 25 mM NaCl, 10 mM HEPES, and 0.1 mM MgCl$_2$, pH 7.2), the plasma membrane was then selectively permeabilized in the same buffer with 0.01% digitonin for 2 min at 37 °C. Basal ER calcium level was determined by measuring the fluorescence emitted at 510 nm after the cells were excited at 488 nm and normalized by DAPI signal. Later, cells were stimulated by different chemicals (1 mM EGTA, 100 μM IP3) and monitored the change of the fluorescence every 20–30 s for 5–8 min to determine the calcium leakage and release from ER.

**Insulin secretion.** Min6 cells were cultured in 15% FBS in DMEM high glucose medium (Gibco) supplemented with 0.5% β-mercaptoethanol and 1% penicillin–streptomycin (Gibco) and transfected with indicated siRNAs for 72 h. Cell were then incubated with indicated concentrations of glucose or KCl, and then supernatant samples were harvested. Insulin levels in the supernatant samples were assessed by Ultra Sensitive Mouse Insulin ELISA Kit (Crystal Chem, #90080), according to the manufacturer's instructions. Corresponding cell lysates were centrifuged at 12,000 × g for 10 min at 4 °C to clear the lysates. One hundred

microliters of the cleared lysates were dried at room temperature and the pellets were dissolved in 100 μl dH₂O and used for DNA concentration measurement. The rest of 400 μl cleared lysates were diluted 1:1000 and were used for insulin ELISA assay. Finally, secreted insulin values were normalized to the insulin contents of the lysates.

**Statistics**. All the experiments were performed more than three times. Quantitative data are presented as mean ± standard error of means or standard deviation. Student's $t$ tests were used to analyze the significance of difference between different samples.

**Reporting summary**. Further information on research design is available in the Nature Research Reporting Summary linked to this article.

## Data availability

Data supporting the findings of this study are available within the paper and its Supplementary Information files, and are available from the corresponding author upon reasonable request. The source data underlying all figures are provided as Source Data file. Ribo-seq and RNA-seq data are deposited in GEO accession numbers GSE136350 and GSE136334, respectively. The source data underlying Figs. 2–6 and Supplementary Figs. 2, 5, and 6 are provided as a Source data file. Uncropped images of northern and western blots are provided as a Source data file.

## Code availability

We used an in-house R code to analyze tail-sequencing data. The code is previously published[8].

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

## Acknowledgements

This work was supported by grants from the US National Institute of General Medical Sciences (NIH-NIGMS) (R01GM086386) to R.I.G, the National Institute of Diabetes and Digestive and Kidney Diseases (NIH-NIDDK) (K01DK121861) to M.P. and a fellowship from the Manton Center for Orphan Disease Research to M.P. We thank Dr. R. Nishinakamura from Kumamoto University, Japan, for generous sharing of the Osr1-GFP reporter mouse ESC line.

## Author contributions

M.P. performed most of the experiments. Q.L., A.G.E., and M.P. performed bioinformatic analyses. C.-H.W. and Y.-H.T. conceptualized and performed calcium concentration measurements. F.S. performed insulin ELISA. M.P. and R.I.G. designed all the experiments, analyzed data, and wrote the manuscript.

## Competing interests

R.I.G. is co-founder and scientific advisory board member of 28-7 Therapeutics. The remaining authors declare no competing interests.
