## [Peer Review File · Nature Communications]

Reviewers' comments:

Reviewer #1 (Remarks to the Author):

The group of Richard Gregory was a first to discover the role of DIS3L2 exonuclease and RNA 3'-end oligouridylation in a cytoplasmic RNA metabolism in human cells (Chang et al., Nature 2013). DIS3L2 malfunction had been earlier linked to a rare disease of the childhood, called Perlman syndrome, which affects mostly kidneys and increases the risk of developing Wilm's tumor, however the molecular mechanism underlying pathogenesis of this disorder has remained unknown. Since their first publication on DIS3L2, R.I. Gregory and co-workers have documented the involvement of DIS3L2-mediated decay pathway (DMD) in the quality control of multiple non-coding RNA species, including pre-miRNA, RMRP RNA, 7SL component of signal recognition particle (SRP) (Pirouz et al., 2016). These findings have been recapitulated and extended to other ncRNA classes by many other research groups and have been shown to be conserved across eukaryotic lineage, with the exception of budding yeast, which do not have DIS3L2 homolog. Nonetheless, the key DIS3L2 RNA target in the context of Perlman syndrome has not been characterized to date.

I would say that the present manuscript by Pirouz and colleagues is truly novel and ground-breaking in that it addresses for the first time the molecular basis of the Perlman syndrome. I have no doubts that it will be interesting to a wide community of researchers, because it touches many important aspects of cell physiology, including – apart from ncRNA metabolism – control of ER-associated translation, maintenance of calcium homeostasis as well as differentiation of stem cells. Therefore, I believe it fully meets the criteria of publication in Nature Communications.

I am impressed by the exceptionally logical sequence of experiments, which make a coherent story, whereby DIS3L2 dysfunction leads to accumulation of aberrant 3'-oligouridylated 7SL RNA species, causing formation of non-functional SRP and this in turn results in the loss of ER-mediated translation and calcium leakage from the ER to the cytoplasm. This cascade of events entails deregulation of cell differentiation, which likely explains kidney overgrowth as one of the Perlman syndrome symptoms. The novelty of these results is indisputable and represents a significant steps towards our understanding why the disorder develops this specific way.

All performed experiments, which required substantial expertise in up-to-date molecular and cell biology methods, are very well controlled and verified by mutually complementary techniques. With few little exceptions, which rather do not apply to key experiments from which the most interesting conclusions have been drawn, I assess the quality of the data presented in the manuscript as very high.

Importantly, contrary to previous reports on DIS3L2 substrates, which utilized RNA-seq for DIS3L2 mutant lines or depleted cells, or variations of CLIP high-throughput experiments, Pirouz et al. employed ribosome profiling on mESCs from DIS3L2 knock-out mice to assess how translation of particular mRNAs is affected by DIS3L2 dysfunction.

Remark 1: The authors emphasize that, in agreement with previous studies, global mRNA expression levels are not changed. However, there were some indications that transcripts coding for replication-dependent histones might be direct targets of DIS3L2. Could they comment on how these mRNAs behaved in their analyses?

Ribo-seq approach revealed both translationally down- and upregulated transcripts. The authors next inspected the former class in detail, unveiling that RPFs distribution across coding sequences tends to decrease, albeit there is a specific accumulation of ribosome protected fragments in the vicinity of the CDS 5'-termini.

Remark 2: Could the authors elaborate a bit more on the possible mechanism of this phenomenon? Why is the translation impaired? Could be some ribosome stalling involved?

GO analysis identified many membrane- and ER-localized proteins encoded by translationally downregulated genes, suggesting that ER-associated translation might be impaired upon DIS3L2 dysfunction. This was confirmed by metabolic labeling of newly synthesized proteins with radioactive methionine, which revealed decreased production of proteins secreted to the media (but not intracellular proteins). Furthermore, thapsigargin-mediated inhibition of ER-associated translation impaired de novo protein synthesis more robustly in DIS3L2 knock-out than in control cells.

Remark 3: The graphs and statistics are generally quite convincing, but I reckon it would be nice to see some SDS-PAGE gel with equal protein amounts loaded and its exposure against phosphoscreen to visualize differences in de novo secreted proteins' synthesis efficiency between DIS3L2 KO and controls. Moreover, could some later time-point that 4h be included in the analysis – the differences would be probably more pronounced.

A more direct proof supporting the role of DIS3L2 in proper ER-mediated translation of secreted proteins was obtained in elegant experiments using secreted and non-secreted luciferase reporters in both mESC KO cells and in various human cell lines treated with siRNAs against DIS3L2. Importantly, rescue experiments with WT DIS3L2, but not its catalytically dead variant restored the wild-type levels of secreted luciferase in KO cells, showing that the enzymatic activity of the protein is indispensable for ER-associated translation.

Remark 4: Why are the levels of exogenous, FLAG-tagged DIS3L2 proteins so much higher in control than in KO cells (Fig. 2D, upper panel)? Although it will probably not influence the conclusions, since the levels of WT and D389N mutant are comparable within either control or KO group, could the authors repeat the experiment so that the amounts of exogenously expressed DIS3L2 variants are more similar in control and KO cells?

Remark 5: There is a clear technical problem with loading control (anti-tubulin antibodies) in the Fig. 2E (upper panel), which should be eliminated in the revised version of the manuscript.

Next, Pirouz et al. investigated the cause of reduced ER-associated translation taking into account DIS3L2 targets in the cell. Their first guess was possible general impairment of the protein translation machinery due to ribosome malfunction or defective tRNAs. However, no change in mature rRNA levels or polysome profiles could be detected in DIS3L2 KO cells. Likewise, codon usage frequencies were unaffected, demonstrating that codon readout by (probably) defective tRNAs entering the ribosomes is rather unchanged.

Remark 6: To make the entire picture more complete, results of rRNA and tRNA analysis in the form of northern-blots which would demonstrate their misprocessing upon DIS3L2 loss should be shown in the Supplementary data, along with comparison of polysome profiles for KO and control cells.

The next series of experiments is dedicated to showing the involvement of DMD pathway in the surveillance of 7SL RNA, which appears to be the key DIS3L2 target in the context of correct ER-associated translation. Several lines of evidence strongly support this conclusion. First, aberrantly truncated, oligouridylated 7SL RNA species bound to DIS3L2 mutant accumulated in DIS3L2 KO cells in RIP experiment. Co-IP analyses with antibodies against SRP component and ribosomal RPL23a in conjunction with polysome profile examination showed that these incorrect 7SL transcripts are present within SRP and associate with translationally active ribosomes. Defect of ER-associated translation resulting from abnormal SRP synthesis could be reversed by siRNA-mediated depletion of TUTases 4 and 7, documenting also the role of these enzymes in uridylation-dependent 7SL RNA surveillance. Eventually, outcome of DIS3L2 defect was mimicked by overexpression of truncated 7SL RNA, which inhibited ER-associated translation of secreted luciferase reporter.

While the data described so far have implications mainly for the field of RNA biology, their consequences are much broader. ER is not only the organelle important for protein folding and trafficking, but also a crucial reservoir of calcium. Accordingly, the authors demonstrate that DIS3L2 KO cells display increased efflux of calcium ions from the ER to the cytosol. This disturbed calcium homeostasis is synergistically increased by treatment of cells with reagents stimulating various calcium channels in ER membrane.

Since calcium signaling affects cell differentiation and organ development, the observations made so far prompted the authors to check these phenomena in the context of Perlman syndrome pathogenesis. To this end, they utilized a cell line expressing renal-specific Osr1-GFP reporter, which was subjected to RNAi using shRNA targeting DIS3L2 or lacZ (negative control).

Remark 7: Most of the results presented until this point were based on DIS3L2 knock-out. I believe it needs to be explained why shRNA-mediated DIS3L2 depletion was used for the most impactful experiments presented in the manuscript. Wouldn't it be better to express Osr1-GFP reporter in the DIS3L2 KO cell line instead? The authors should consider repeating at least some experiments on differentiation employing such experimental setup, which would be more complementary to other data.

While pluripotency of such mESCs was unaffected by DIS3L2 downregulation, spontaneous differentiation of these cells led to the formation of larger embryoid bodies (EBs) than in control cells. Poly(A)⁺ RNA-seq revealed significant changes in gene expression, particularly at later steps of EBs differentiation. Spectacularly, DEGs included genes coding for factors involved in calcium homeostasis, ECM organization as well as those associated with phenotypes characteristic for Perlman syndrome patients. Moreover, deregulation of renal-associated TFs accompanied ESCs differentiation. Accordingly, number of cells expressing Osr1-GFP reporter and renal-specific markers was increased during differentiation towards renal lineage, which is in line with Perlman syndrome manifestation in the form of kidney overgrowth and tendency of Wilms' tumor development.

Since this is the first such a comprehensive analysis of DIS3L2 role in the pathogenesis of Perlman syndrome and the experiments are very well planned and executed, the manuscript definitely deserves publication in Nature Communications provided that several small criticisms mentioned above will be addressed during revision phase.

Minor comments:

- throughout the manuscript, DIS3L2 protein name should be written with capital letters (as is in the list of key words at page 1);
- There is still a mess with naming TUTases, however the most recent recommendations is to use names TENT3A and TENT3B for TUTase 4 and 7, respectively – this should be mentioned in the first paragraph of the Introduction at page 3 (line 6);
- If authors decide to use the word “uridylyl transferase”, then they should also consequently change the term “uridylylated” into “uridylylated”; otherwise. “uridylyl transferase” should be changed into “uridyl transferase”;
- “poly-Uridine” should be changed into “poly-uridine” or “polyuridine” (page 3, line 11);
- “abberant” should be changed into “aberrant” (page 4, line 10 and page 10, line 8);
- “Inhibition of ER-mediated translation inhibition by thapsigargin” should be changed into “Inhibition of ER-mediated translation by thapsigargin” (page 5, line 8);
- “in vitro” should be italicized (page 7, last line) – this also refers to other instances in the text;
- “Dis3L2-defiecient” should be corrected into “DIS3L2-deficient” (page 9, line 5);
- “mitochondrial” should be corrected into “mitochondria” (page 10, line 17);
- Authors should decide which abbreviation – q.RT-PCR or qRT-PCR they want to use throughout the manuscript;
- Fig. 5f Legend – “d10” should be changed into “d12”;
- “bacteria” should be corrected into “plasmid” (page 27, line 2);
- “normalization” should be corrected into “normalized” (page 27, line 8);
- “edegR” should be corrected into “edgeR” (page 29, line 18);
- “disitonin” should be corrected into “digitonin” (page 31, line 2).

Reviewer #2 (Remarks to the Author):

In “The Perlman Syndrome Dis312 exonuclease safeguard ER-associated translation, Ca²⁺ homeostasis, and stem cell differentiation,” Richard Gregory and colleagues aim to determine the functionally most important target(s) of Dis312, a protein known to degrade multiple classes of non-coding RNAs when their 3' ends are oligouridylated. They present data that suggest that the specific failure to degrade the critical RNA component of the signal recognition particle (SRP), 7SL, leads to decreased translation of proteins at the ER, which in turn leads to defects in Ca homeostasis and ES differentiation. The basic idea is the retention of truncated and uridylated 7SL gums up the ability to translate transcripts that encode a signal peptide. The paper presents a number of experiments to support the model. The connections between each step though (accumulation of 7SL, reduced translation, calcium homeostasis, and stem cell differentiation) represent rather large leaps. For example, it is unclear how uridylated 7SL would gum up the system, why only a subset of ER translated proteins are affected, or how defects in ER proteins would lead to specific differentiation defects. Presumably these questions are being saved for another day. The findings should be of interest to readers of NC, especially those interested in RNA surveillance. However, a number of details surrounding their experiments need to be addressed first before being suitable for publication.

Figure 1. 1a: According to methods, cutoff is set at 1.5x, with no statistical measure. Authors should include a statistical cutoff and clearly state cutoffs in figure and/or legend. 1b: Again, need to present some statistical argument that the difference is meaningful. 1c: Would like to see same for up genes. 1d: again, would like to see broken down into down, unchanged, and up. Importantly, there needs to be a discussion of why only a small fraction of ER translated proteins are affected.

Figure 2. 2b/c: not sure what the addition of thapsigargin shows. The effect is small, non-specific to KO in case of the Gluc, and any effect is likely very indirect. I recommend removing. Also, in C, it looks like downregulation is as at translation level rather than translocation across ER (which is the presumed function of SRP). Would be helpful to have some discussion of why a defect in SRP, which directs transcript to ER, is inhibiting translation. This is similar to comment above, i.e. how does uridylated/truncated 7SL gum up translation of signal peptide containing proteins.

Figure 3. 3a,d,e: From these data, it is unclear what fraction of total 7SL transcripts are uridylated/truncated in each experiment. This number is important. If it is really small (say less than 5%), it is unclear how it could be gumming up the entire system. 3c: Needs much better explanation. Not sure it adds much. 3f: not sure what the point of this panel is. Is the idea that the uridylated 7SL is enriched in the translated fraction? If so, need to show data for monosome fraction. 3h: Given that the overall increase in 7SL expression is minimal (fig. S2C), it is difficult to understand the basis for the results. Is there something special about expressing mouse 7SL in human cells that even a small overall (human and mouse) increase can rescue?

Figure 4. minor issue: very difficult to read fig. 4b.

Figure 5. 5c: Why are the DEGs almost entirely upregulated genes, especially d12? 5d: In contrast to text I would conclude from this clustering that it is embryonic day not genotype that is predominantly

separating samples. A principal component analysis might help to see if there is a principal component that nicely separates genotype and determine what genes are most heavily weighted in that PC. 5g: not sure how to interpret this analysis or what value it adds.

Figure 6. 6a,b: for this data, would like to see fold change in addition to z score. That is what is the actual change in expression. If small changes with minimal variance are underlying the z score, that would be less meaningful. 6d-f: I appreciate what authors are trying to do, but these results are a very far stretch from describing the disease phenotype associated with the kidney. The interpretation is overstated and oversimplified given the minimal data.

Reviewer #3 (Remarks to the Author):

Pirouz et al. analyze the consequences of removing Dis3l2 function in mammalian cells. Based on ribosome profiling, they report impaired translation of secretory and membrane proteins and claim a mechanistic link to impaired quality control of the 7SL RNA component of the signal recognition particle (SRP) that targets most proteins to the endoplasmic reticulum (ER) in mammalian cells. The authors try to link these findings to ER calcium leakage and dysregulated ESC differentiation. The experiments are well-executed and the independent observations of (i) different translation efficiencies, (ii) aberrant SRP 7SL RNA, (iii) calcium dysregulation, and (iv) differentiation phenotypes are clear. These results should be useful towards understanding the role of Dis3l2-mediated RNA decay in physiology. However, the suggested causality between Dis3l2 function, protein synthesis at the ER, and the proposed downstream “SRPopathy” phenotypes are weak and overinterpreted in this manuscript as written. At minimum, the text implying these links should be significantly toned down before publication.

Major comments:

1. Although the data showing aberrant SRP 7SL RNA in Dis3l2 KO cells (Fig. 3) are clear, the evidence linking Dis3l2 function and impaired “ER-mediated translation” is not convincing and requires more controls and/or reinterpretation:

a. In Fig. 1, if predicted ER-destined proteins make up at most 20% of translationally downregulated genes in KO cells (Fig. 1d), can a difference in ribosome occupancy by metagene analysis of the entire set of downregulated genes (Fig. 1b) be linked specifically to ER-associated translation?

Do downregulated non-ER transcripts show the same RPF pattern in the KO cells? How do downregulated non-ER transcripts compared to unaffected non-ER transcripts? More results are needed to demonstrate that Dis3l2 KO is not inducing a more general translational repression mechanism.

b. Is there a Dis3l2-independent difference of ribosomal occupancy on ER-destined transcripts vs. others? What happens if the same analysis shown in Supplemental Fig. 1c is done for control cells?

c. Fig. 1d shows some preference of predicted ER proteins in the translationally downregulated genes, but these are still only ~1/8 of all predicted ER-targeted proteins. What makes these clients particularly sensitive to Dis3l2 KO? Are SRP clients de-enriched in the translationally up-regulated genes in the KO cells?

d. I am not convinced that the metabolic labeling experiment in Fig. 2a is a fair assessment of secreted vs. non-secreted protein synthesis. It seems that the lysate and supernatant fractions were collected in separate experiments. In addition, while the protein content of the lysates was normalized before scintillation counting, that of the supernatant was not addressed, allowing for far more variability that may arise not only from translation and secretion efficiency, but also different rates of cell growth and amino acid metabolism. It's also not clear if the radioactivity measured in the supernatant is truly incorporated into proteins vs. free radiolabeled methionine. A more equal comparison would be to start with the same number of WT and KO cells, collect the supernatant and lysate in the same experiment, and remove free amino acids from each sample.

e. Many experiments examining ER-mediated translation are done with a secreted luciferase reporter. To rule out SRP-independent effects, controls should be included showing the level of mRNA expression of this reporter in the conditions analyzed in Fig. 2c-2e, especially since the secreted and cytosolic reporters are in different plasmid backbones, encode different enzymes, and are expressed by transient transfection, which is additional subject to variable transfection efficiencies and cell-to-cell heterogeneity.

f. Stronger support for a link between Dis3l2 function and SRP client translation could be achieved by biochemically analyzing endogenous Dis3l2-dependent and independent clients by metabolic labeling and immunoprecipitations in control and KO cells with and without Dis3l2 re-expression. Alternatively, the authors could use matched reporter constructs that differ in the presence of an ER-targeting signal sequence, or compared to an SRP-independent ER targeting signal, such as a C-terminal tail-anchored transmembrane anchor.

g. Can the authors comment on the mechanism by which impaired SRP function in Dis3l2 KO cells leads to reduced translation, rather than protein mislocalization as observed in Costa et al. (2018), ref. 39?

2. The authors suggest a causal link between translational downregulation of ER-destined proteins and calcium leakage. While the data independently showing translational downregulation of some genes and calcium dysregulation in Dis3l2 cells are clear, there is no evidence supporting the claim that the first causes the second. As an example, thapsigargin treatment causes cytosolic calcium accumulation, which also leads to impaired protein secretion in Dis3l2 KO cells (Fig. 2b and 2c). This already suggests that the two observations of impaired protein secretion/translation and calcium dysregulation cannot be cleanly uncoupled in these experiments.

3. Similarly, while the experiments looking at cellular differentiation are beautiful, they remain observational. Descriptions suggesting causal links should be avoided in the text. Alternatively, the hypothesis should be directly tested. For example, rescuing the differentiation phenotype by re-expressing wildtype SRP 7SL RNA would give more credence to an "SRPopathy" model. As it stands, there is no data in the paper to rule out other plausible hypotheses. Perhaps Dis3l2 KO have misregulated miRNA targets that lead to the observed phenotypes. It also remains unclear why only a minority of predicted SRP clients are affected in an "SRPopathy" (see point 1b).

Minor points:

1. The authors demonstrate their expertise in RNA biology and metabolism in the first paragraph of the introduction. In contrast, the second paragraph describing SRP function was vague, lacking specific information that could influence interpretations in the manuscript. The general mechanism of SRP-

mediated targeting of ribosomes synthesizing secretory and membrane proteins to the ER, as well as the scope and characteristics of SRP clients, are well studied and included in most college-level biology textbooks, such as *Molecular Biology of the Cell* by Alberts et al. It would be helpful if the relevant parts of this established work are introduced to help place the authors' findings in context.

2. The term/concept of "ER-mediated translation" is not clear to me. It implies that ribosomes at the ER translate more efficiently. Is this what the authors mean (and if so, what is the evidence for this)? Or does it refer only to protein synthesis at the ER?

3. Please be wary of the term "protein translation". mRNAs are translated, proteins are not. Alternatives are "translation" or "protein synthesis".

4. "aberrant" is misspelled as "abberant" throughout the text.

Reviewers' comments:

Reviewer #1 (Remarks to the Author):

The group of Richard Gregory was a first to discover the role of DIS3L2 exonuclease and RNA 3'-end oligouridylation in a cytoplasmic RNA metabolism in human cells (Chang et al., Nature 2013). DIS3L2 malfunction had been earlier linked to a rare disease of the childhood, called Perlman syndrome, which affects mostly kidneys and increases the risk of developing Wilm's tumor, however the molecular mechanism underlying pathogenesis of this disorder has remained unknown. Since their first publication on DIS3L2, R.I. Gregory and co-workers have documented the involvement of DIS3L2-mediated decay pathway (DMD) in the quality control of multiple non-coding RNA species, including pre-miRNA, RMRP RNA, 7SL component of signal recognition particle (SRP) (Pirouz et al., 2016). These findings have been recapitulated and extended to other ncRNA classes by many other research groups and have been shown to be conserved across eukaryotic lineage, with the exception of budding yeast, which do not have DIS3L2 homolog. Nonetheless, the key DIS3L2 RNA target in the context of Perlman syndrome has not been characterized to date.

I would say that the present manuscript by Pirouz and colleagues is truly novel and ground-breaking in that it addresses for the first time the molecular basis of the Perlman syndrome. I have no doubts that it will be interesting to a wide community of researchers, because it touches many important aspects of cell physiology, including – apart from ncRNA metabolism – control of ER-associated translation, maintenance of calcium homeostasis as well as differentiation of stem cells. Therefore, I believe it fully meets the criteria of publication in Nature Communications.

I am impressed by the exceptionally logical sequence of experiments, which make a coherent story, whereby DIS3L2 dysfunction leads to accumulation of aberrant 3'-oligouridylated 7SL RNA species, causing formation of non-functional SRP and this in turn results in the loss of ER-mediated translation and calcium leakage from the ER to the cytoplasm. This cascade of events entails deregulation of cell differentiation, which likely explains kidney overgrowth as one of the Perlman syndrome symptoms. The novelty of these results is indisputable and represents a significant steps towards our understanding why the disorder develops this specific way.

All performed experiments, which required substantial expertise in up-to-date molecular and cell biology methods, are very well controlled and verified by mutually complementary techniques. With few little exceptions, which rather do not apply to key experiments from which the most interesting conclusions have been drawn, I assess the quality of the data presented in the manuscript as very high.

Importantly, contrary to previous reports on DIS3L2 substrates, which utilized RNA-seq for DIS3L2 mutant lines or depleted cells, or variations of CLIP high-throughput experiments, Pirouz et al. employed ribosome profiling on mESCs from DIS3L2 knock-out mice to assess how translation of particular mRNAs is affected by DIS3L2 dysfunction.

We thank this reviewer for their very positive and comprehensive feedback on our manuscript. We have performed several different experiments and analyses to address their constructive remarks and further support our conclusions.

Remark 1: The authors emphasize that, in agreement with previous studies, global mRNA expression levels are not changed. However, there were some indications that transcripts coding for replication-dependent histones might be direct targets of DIS3L2. Could they comment on how these mRNAs behaved in their analyses?

We checked expression of replication-dependent histone mRNAs and found no global dysregulation of these mRNAs in DIS3L2 knockout mESCs. Among the reported DIS3L2 targets, HIST1H1E, HIST1H2AG, HIST1H2BN, HIST1H3B, HIST1H3D, and HIST1H4A show no change in mRNA levels nor their translation efficiencies; HIST1H2AL, HIST1H2AM, HIST1H2BD, HIST1H2BO were not detectably expressed in mESCs; and HIST1H4I showed a modest decrease (as opposed to any increase) in DIS3L2 knockout mESCs (Supplementary Table 1). Altogether, we believe that DIS3L2 deficiency leads to minimal changes in the steady state expression level of histone mRNAs in mESCs.

Remark 2: Could the authors elaborate a bit more on the possible mechanism of this phenomenon? Why is the translation impaired? Could be some ribosome stalling involved?

As we have shown in Figures 2-3, the translation impairment is due to defective 7SL RNA processing, that leads to SRP complex dysfunction in DIS3L2 knockout cells. RPF distribution of translationally downregulated mRNAs (Figure 1d) indicates that this phenomenon is likely due to a stalled ribosomes on ER-targeted mRNAs. These points are more explicitly discussed in the revised manuscript.

Remark 3: The graphs and statistics are generally quite convincing, but I reckon it would be nice to see some SDS-PAGE gel with equal protein amounts loaded and its exposure against phosphoscreen to visualize differences in de novo secreted proteins' synthesis efficiency between DIS3L2 KO and controls. Moreover, could some later time-point that 4h be included in the analysis – the differences would be probably more pronounced.

We have repeated this experiment and included a later time point (8h) as suggested by the reviewer in the revised manuscript. As anticipated, the difference in the secretion of newly synthesized proteins is more pronounced at later time points (New Figure 2a). Moreover, we ran SDS-PAGE separation of cellular lysates and supernatant samples and performed autoradiography of radioactive S³⁵. However, the signal intensity of the supernatant samples is very low and could not be detected even after a 1-month exposure (please see the image below). Notably, the scintillation machine values of the signal intensity from supernatants is almost 1/100th of their corresponding lysate samples, likely explaining this observation. We believe that, in contrast to the scintillation machine, autoradiography of the supernatant samples on SDS-PAGE gels has a technical limitation and may not be sensitive enough to detect radiolabeled secretome from the large volume of cell culture media.

Remark 4: Why are the levels of exogenous, FLAG-tagged DIS3L2 proteins so much higher in control than in KO cells (Fig. 2D, upper panel)? Although it will probably not influence the conclusions, since the levels of WT and D389N mutant are comparable within either control or KO group, could the authors repeat the experiment so that the amounts of exogenously expressed DIS3L2 variants are more similar in control and KO cells?

For reasons that are unclear we have observed that the level of exogenously-expressed DIS3L2 is typically lower in knockout cells than WT cells (Pirouz et al., 2016; Pirouz et al., 2019). However, we repeated the experiment and now present data with a more similar level of FLAG-DIS3L2 protein expression in transfected control and knockout cells (New Figure 2d and New Supplementary Figure 2c). Nevertheless, as the reviewer indicated this did not change the main conclusions as the levels of WT and D389N mutant are comparable within either control or KO group.

Remark 5: There is a clear technical problem with loading control (anti-tubulin antibodies) in the Fig. 2E (upper panel), which should be eliminated in the revised version of the manuscript.

A new WB analysis (New Figure 2e) is provided in the revised manuscript using anti beta-actin antibody for the loading control.

Remark 6: To make the entire picture more complete, results of rRNA and tRNA analysis in the form of northern-blot which would demonstrate their misprocessing upon DIS3L2 loss should be shown in the Supplementary data, along with comparison of polysome profiles for KO and control cells.

A role for DIS3L2 in the surveillance of mis-processed rRNAs has been comprehensively studied (including Northern blots) and recently published by our group (Pirouz et al., NSMB, 2019). While we cannot entirely rule out the pathological roles of mis-processed rRNAs in DIS3L2 knockout cells this should be expected to impact global translation. However our Ribo-Seq, metabolic labeling, and reporter experiments (Figures 1 and 2) clearly support a more specific defect in ER-mediated mRNA translation. Moreover, reporter experiments more directly implicate 7SL RNA where we found that ectopic expression of WT 7SL RNA can rescue the defective ER-mediated translation of the reporter, whereas expression of truncated 7SL RNA further compromised ER-mediated translation of the secreted luciferase reporter. To examine the possible involvement of tRNA dysregulation, we performed a more detailed analysis of codon occupancies in E-, P- and A-sites of ribosomes (New Supplementary Figures 3a,b) as well as codon frequencies in differentially translated mRNAs (New Supplementary Figure 4a) and transcripts with or without signal peptide sequences (New Supplementary Figure 4b). These results show no differential codon occupancies between DIS3L2-targeted and -nontargeted tRNAs. Finally, for the requested polysome profiles please see (Pirouz et al., NSMB, 2019), where no obvious difference was observed in the profiles of control or DIS3L2 KO ESCs. Together these results support that rRNA or tRNA are unlikely to be responsible for the altered mRNA translation observed in DIS3L2-deficient cells.

Remark 7: Most of the results presented until this point were based on DIS3L2 knock-out. I believe it needs to be explained why shRNA-mediated DIS3L2 depletion was used for the most impactful experiments presented in the manuscript. Wouldn't it be better to express Osr1-GFP reporter in the DIS3L2 KO cell line instead? The authors should consider repeating at least some experiments on differentiation employing such experimental setup, which would be more complementary to other data.

Originally, we obtained the published Osr1-GFP reporter-expressing mESC line from another lab, in which we depleted DIS3L2 using specific shRNAs. We did not have access to the Osr1-GFP reporter plasmid itself to express it in control vs. DIS3L2 knockout cell lines. Nevertheless, the specific DIS3L2 shRNA that we used is very effective in downregulating DIS3L2 expression as shown by RT-qPCR, and WB (Supplementary Figure 6a) as well as a consequent inhibition of ER-reporters (Supplementary Figure 6b), similar to the behavior of DIS3L2 knockout cells. Thus, we believe the shRNA-mediated DIS3L2 knockdown is comparably efficient in perturbing mRNA translation and consequently ESC differentiation. In the revised manuscript we verified the findings from RNA-seq with new qRT-PCR analysis of renal-specific markers and showed upregulation of these markers in shDIS3L2-derived metanephric mesenchyme cells compared to the controls (New Figure 6g). This result extends our analysis beyond the previously observed increased number of Osr1-GFP cells to other renal markers. Also, to experimentally address the reviewer's comment, as well as to address the pathological effects of misprocessed (truncated) 7SL RNA on renal differentiation, we utilized our control vs. knockout mESC lines (without Osr1-GFP reporter) that were transduced with lentiviruses to stably express a pLK0.1-based empty vector (as control) or truncated 7SL RNA from the endogenous 7SL RNA promoter sequence. qRT-PCR analysis of differentiated cells showed that: 1) similar to shDIS3L2 cells, DIS3L2 knockout cells also show elevated levels of renal-specific markers Six2 and IGF2 as well as calcium-sensing factor Camk1g (New Figure 6h) and more interestingly, 2) over-expression of truncated 7SL RNA further enhanced the expression of these markers only in DIS3L2 knockout cells, but not in the control cells that express functional DIS3L2 protein that would recognize and degrade truncated 7SL RNAs. These results underscore the potential negative effects of truncated 7SL RNA species on renal differentiation and highlights the specific function of DIS3L2 in their elimination.

Since this is the first such a comprehensive analysis of DIS3L2 role in the pathogenesis of Perlman syndrome and the experiments are very well planned and executed, the manuscript definitely deserves publication in Nature Communications provided that several small criticisms mentioned above will be addressed during revision phase.

We are grateful to the reviewer for supporting publication of our work.

Minor comments:

- throughout the manuscript, DIS3L2 protein name should be written with capital letters (as is in the list of key words at page 1);
We have made this change to the text and figure labels.
- There is still a mess with naming TUTases, however the most recent recommendations is to use names TENT3A and TENT3B for TUTase 4 and 7, respectively – this should be mentioned in the first paragraph of the Introduction at page 3 (line 6);
We have made this change to the text.
- If authors decide to use the word “uridylyl transferase”, then they should also consequently change the term “uridylated” into “uridylylated”; otherwise, “uridylyl transferase” should be changed into “uridyl transferase”;
We have made this change to the text and refer to ‘uridylyl’ throughout the manuscript.
- “poly-Uridine” should be changed into “poly-uridine” or “polyuridine” (page 3, line 11); *We have made this change to the text and refer to ‘poly-uridine’ throughout the manuscript.*
- “abberant” should be changed into “aberrant” (page 4, line 10 and page 10, line 8); *This is now corrected in the revised manuscript.*
- “Inhibition of ER-mediated translation inhibition by thapsigargin” should be changed into “Inhibition of ER-mediated translation by thapsigargin” (page 5, line 8);

This is now corrected in the revised manuscript.

- “in vitro” should be italicized (page 7, last line) – this also refers to other instances in the text; *This is now corrected in throughout the revised manuscript.*
- “Dis3L2-deficient” should be corrected into “DIS3L2-deficient” (page 9, line 5); *This is now corrected in the revised manuscript.*
- “mitochondrial” should be corrected into “mitochondria” (page 10, line 17); *This is now corrected in the revised manuscript.*
- Authors should decide which abbreviation – q.RT-PCR or qRT-PCR they want to use throughout the manuscript; *This is now changed to qRT-PCR throughout the revised text.*
- Fig. 5f Legend – “d10” should be changed into “d12”; *This is now corrected in the revised manuscript.*
- “bacteria” should be corrected into “plasmid” (page 27, line 2); *This is now corrected in the revised manuscript.*
- “normalization” should be corrected into “normalized” (page 27, line 8); *This is now corrected in the revised manuscript.*
- “edegR” should be corrected into “edgeR” (page 29, line 18); *This is now corrected in the revised manuscript.*
- “disitonin” should be corrected into “digitonin” (page 31, line 2). *This is now corrected in the revised manuscript.*

Reviewer #2 (Remarks to the Author):

In “The Perlman Syndrome Dis3L2 exonuclease safeguard ER-associated translation, Ca²⁺ homeostasis, and stem cell differentiation,” Richard Gregory and colleagues aim to determine the functionally most important target(s) of Dis3L2, a protein known to degrade multiple classes of non-coding RNAs when their 3' ends are oligouridylated. They present data that suggest that the specific failure to degrade the critical RNA component of the signal recognition particle (SRP), 7SL, leads to decreased translation of proteins at the ER, which in turns leads to defects in Ca homeostasis and ES differentiation. The basic idea is the retention of truncated and uridylated 7SL gums up the ability to translate transcripts that encode a signal peptide. The paper presents a number of experiments to support the model. The connections between each step though (accumulation of 7SL, reduced translation, calcium homeostasis, and stem cell differentiation) represent rather large leaps. For example, it is unclear how uridylated 7SL would gum up the system, why only a subset of ER translated proteins are affected, or how defects in ER proteins would lead to specific differentiation defects. Presumably these questions are being saved for another day. The findings should be of interest to readers of NC, especially those interested in RNA surveillance. However, a number of details surrounding their experiments need to be addressed first before being suitable for publication.

We are grateful for the positive evaluation and constructive feedback on our original manuscript. In our revised manuscript we have performed several new experiments and data analyses to address the reviewer's specific comments:

Figure 1. 1a: According to methods, cutoff is set at 1.5x, with no statistical measure. Authors should include a statistical cutoff and clearly state cutoffs in figure and/or legend.

We apologize for not being clearer in the original manuscript. In Figure 1a, the cut-off was a 2-fold change (log₂=1) as it was indicated in the original figure and also the revised one. 2-fold change is often used as a meaningful threshold for the majority of gene expression data analyses, including translation efficiencies (Figure

1a). The 2-fold change cutoff is now mentioned in the figure legend and also in the methods section. In the new Figure 1b, we also presented a statistical analysis of transcripts with differential translation efficiencies between control and knockout cells (please also see Supplementary Table 2 for P-values). Here, again a 2-fold change with a P-value ≤ 0.05 is considered as statistically significant.

1b: Again, need to present some statistical argument that the difference is meaningful.

Here (now Figure 1d in the revised manuscript) we show the distribution of RFPs in the translationally downregulated mRNAs. This distribution shows that while there is a decrease in RFPs throughout the body of the mRNA, there is an accumulation of RFPs at the 5' end that likely represents a stalling of Ribosomes. This distribution is clearly distinct from that of mRNAs with increased or unchanged translation efficiency (Supplemental Figure 1b). We consider that these qualitative differences are in support of our model. It is unclear to us exactly what type of statistical test we could perform on this dataset to further highlight these differences.

1c: Would like to see same for up genes.

In the revised manuscript, we also perform GO analysis of translationally upregulated transcripts. This showed a specific enrichment for mitochondrial proteins (New Figure 1c). While this is an interesting observation and merits further investigation, we chose to focus this study on translationally downregulated mRNAs.

1d: again, would like to see broken down into down, unchanged, and up. Importantly, there needs to be a discussion of why only a small fraction of ER translated proteins are affected.

This data is now broken down into down-, unchanged, and up-regulated transcripts (now presented as Revised Figure 1e). Each of these 3 databases (TMHMM, SignalP, and Phobius) cannot themselves accurately predict all of the ER-associated transcripts, but rather each database considers different features to predict mRNAs that are translated at the ER. Nevertheless, according to this new analysis, ER-associated transcripts are significantly more enriched among translationally downregulated transcripts. We also wish to clarify that the analysis in Figure 1e shows the percentage of all translated mRNAs and therefore (depending on the database) about 30-45% of all translated mRNAs are linked to the ER and of these there is a strong and statistically significant decreased translation efficiency (compared to up- or unchanged) using all 3 databases, with between ~30-50% of these mRNAs being downregulated at the translation level in DIS3L2 deficient cells. However, in the revised manuscript we discuss why some but not all ER-associated transcripts are affected by DIS3L2 loss. Briefly, 1) according to previously well-documented findings, not all the ER-associated transcripts utilize SRP-mediated mechanism(s) for their translation; and 2) a small fraction but not all of 7SL RNAs is misprocessed, uridylylated, and targeted by DIS3L2; 3) a subset of mRNAs might be especially sensitive to SRP perturbation, which altogether could explain why not all the ER-mediated translation is shut down upon DIS3L2 loss. These points are now discussed in the revised manuscript.

Figure 2. 2b/c: not sure what the addition of thapsigargin shows. The effect is small, non-specific to KO in case of the Gluc, and any effect is likely very indirect. I recommend removing.

Thapsigargin inhibits ER-mediated translation. Originally, we included these results to show that DIS3L2 knockout cells are more sensitive to ER translation perturbation. Per reviewer's suggestion, and to simplify our conclusions, we removed these data in the revised manuscript.

Also, in C, it looks like downregulation is as at translation level rather than translocation across ER (which is the presumed function of SRP). Would be helpful to have some discussion of why a defect in SRP, which directs transcript to ER, is inhibiting translation. This is similar to comment above, i.e. how does uridylylated/truncated 7SL gum up translation of signal peptide containing proteins.

Respectfully, we did not claim that the defect is at the translocation step across the ER. Also SRP's function is not only to translocate proteins across the ER. As we indicated in the original manuscript and also now highlight more explicitly in the revised manuscript, SRP functions to recognize some ER-associated peptides being translated by free cytosolic ribosomes. SRP, in which 7SL RNA is a critical component, binding to ribosomes is critical to stall translation of the respective mRNAs and resume translation after docking to the ER-membrane. Accordingly, in DIS3L2 knockout cells we see accumulation of RFPs at the 5' end of the translationally downregulated mRNAs (Figure 1d), and translation of the ER-specific luciferase reporter is perturbed not only in the secreted fraction (which could indicate either translation or translocation defects), but also in the lysate sample, the latter strongly supporting the notion that DIS3L2 safeguards ER-mediated mRNA translation. These and additional points are now more thoroughly discussed in the revised manuscript.

Figure 3. 3a,d,e: From these data, it is unclear what fraction of total 7SL transcripts are uridylated/truncated in each experiment. This number is important. If it is really small (say less than 5%), it is unclear how it could be gumming up the entire system.

We share the reviewer's curiosity as to exactly how the accumulation of truncated/extended and uridylated 7SL might inhibit SRP function. We intend to follow up in future to address this question in detail and this level of understanding is reasonably beyond the scope of our original findings reported in this manuscript. The percentage of uridylated 7SL RNAs in DIS3L2 IP sample is provided in Figure 3a (please see the right Y-axis). Accordingly, the majority of DIS3L2-bound 7SL RNAs are uridylated as is expected from previous biochemical experiments and the published DIS3L2-oligoU RNA co-crystal structure. However, this only shows the extent of uridylated (and truncated/extended) 7SL that is associated with DIS3L2 and not in the total pool of cellular 7SL. To gain more insight into the proportion of uridylated 7SL RNAs that accumulate in the absence of DIS3L2, we performed a 7SL northern blot on RNA from control and DIS3L2 KO cells (New Figure 3f). As anticipated by the reviewer this shows that only a small fraction (<10%) of 7SL is elongated (representing oligoU-tailed aberrant 7SL RNA). Although we did not suggest that these aberrant 7SL transcripts 'gum up the entire system' they do cause around 30-40% reduction in the translation of ER-specific luciferase reporters (Figures 2c and f), and lead to decreased translation of ~30-50% of endogenous mRNAs that are predicted to require ER-mediated translation (Figure 1e). Our interpretation of these data is that once accumulated in the cells uridylated 7SL RNA albeit comprising <10% of the total 7SL RNAs in the cells, may have dominant negative effect on SRP complex function and thereby becomes pathological even at low cellular abundance. Consistent with this notion, we find that even a very modest level of 7SL expression (Supplementary Figure S5d, right panel) can have relatively strong effect on reporter gene expression (Figure 3i) and ESC differentiation (New Figure 6h). Finally, we propose that it is an accumulation of aberrant AND uridylated 7SL that is contributing to the observed inhibition of translation at the ER (i.e. not just the truncated 7SL). This is supported by data in Figure 3h where the inhibition of the secreted luciferase reporter is (at least partially) rescued by TUTase knockdown. These points are raised in the revised discussion.

3c: Needs much better explanation. Not sure it adds much.

This is a regression analysis of U-tail probability with respect to the length of truncation or extension of the 7SL RNA in the DIS3L2 RIP sample. Accordingly, there is a positive correlation between the length of the truncation of 7SL RNA and the likelihood that is uridylated. We think this data is informative and supports with a very high statistical power that 7SL RNA truncation induces terminal uridylation.

3f: not sure what the point of this panel is. Is the idea that the uridylated 7SL is enriched in the translated fraction? If so, need to show data for monosome fraction.

The data for monosome fraction is added to the revised manuscript. Accordingly, uridylated 7SL RNA is enriched in the translating ribosomes, especially in the polysomal fractions.

3h: Given that the overall increase in 7SL expression is minimal (fig. S2C), it is difficult to understand the basis for the results. Is there something special about expressing mouse 7SL in human cells that even a small overall (human and mouse) increase can rescue?

We apologize for not being clear in the original manuscript. We used human cells to transfect mouse 7SL RNA expressing vector and check the expression levels of the exogenous vector using mouse-specific primers against 7SL RNA. Accordingly, mouse 7SL RNA is actively expressed in human cells from a construct bearing the endogenous 7SL promoter sequence (previous Figure S2c left panel, new Figure S5d left panel) and by extension in mESC, where the main experiment is performed (Figure 3h). In Figure S5d, right panel, we measured the total expression level of 7SL RNA using mouse/human common primers in the human cells transfected with 7SL RNA expressing plasmid. The results show that 7SL RNA overexpression is still around the physiological levels and not overwhelmingly above the mock-transfected cells (Figure S5d, right panel).

Figure 4. minor issue: very difficult to read fig. 4b.

The graph is replaced with a clearer one in the revised manuscript. The values for "CTRL+Anisomycin" and "Knockout +Anisomycin" (yellow and blue rectangles, respectively) are overlapping and this is why it was not clear in the original version.

Figure 5. 5c: Why are the DEGs almost entirely upregulated genes, especially d12?

This is an interesting observation. According to the gene ontology analyses these transcripts belong to calcium binding proteins, extracellular/secreted factors, or transcription factors associated with progenitor cells and embryonic development. Our hypothesis is that dysregulated calcium homeostasis, and/or a developmental delay in DIS3L2-depleted cells caused mostly an upregulation of gene expression but we do not know why this is associated mostly with increased mRNA expression.

5d: In contrast to text I would conclude from this clustering that it is embryonic day not genotype that is predominantly separating samples. A principal component analysis might help to see if there is a principal component that nicely separates genotype and determine what genes are most heavily weighted in that PC.

It is true that differentiation timepoints heavily contribute to separation of samples and this is because differentiated cells are drastically different from undifferentiated ones no matter which genotypes are considered. A PCA analysis is added to the revised manuscript (New Figure 5e) and accordingly (and similar to the dendrogram provided in Figure 5d) it shows separation of shDIS3L2 or shLacZ samples mainly towards the end of the differentiation time course (d8-d12). Thus, whereas early differentiation is similar in the presence or absence of DIS3L2, the later differentiation is perturbed in the samples depleted of DIS3L2. This is a recurrent theme in different analyses (Figure 5d, 5e, and heatmap data in Figure 5g). The gene ontology analysis of DEG contributing to the PC are depicted in Figure 5h.

5g: not sure how to interpret this analysis or what value it adds.

The gene ontology analysis of DEG contributing to the PC are depicted in Figure 5h (previously, Fig. 5g). These data show that at d12 of differentiation, DEGs between shLacZ and shDIS3L2 samples are associated with different biological processes and molecular functions, possibly explaining various features of Perlman syndrome (upon request we could move these findings to the supplementary figures).

Figure 6. 6a,b: for this data, would like to see fold change in addition to z score. That is what is the actual change in expression. If small changes with minimal variance are underlying the z score, that would be less meaningful.

The actual fold changes for these transcripts are now provided separately in Supplementary Table 4. For several transcripts, it is clear that huge changes occur in their expression during renal differentiation and especially in DIS3L2-depleted samples.

6d-f: I appreciate what authors are trying to do, but these results are a very far stretch from describing the disease phenotype associated with the kidney. The interpretation is overstated and oversimplified given the minimal data.

We have revised the text to avoid overstating and toned down the proposed links between the cellular phenotypes we observe and the kidney phenotypes of Perlman patients. In the revised manuscript, we add additional data supporting that DIS3L2 loss causes renal differentiation defects including the gene expression analysis of renal progenitor markers Six2, Lim1, and IGF2 by qRT-PCR (new Figure 6g). We repeated the renal differentiation experiment in DIS3L2 knockout and control cells with and without ectopic expression of truncated 7SL RNA. These new data show that 1) similar to DIS3L2 knockdown cells, DIS3L2 knockout cells also express higher levels of renal progenitor markers; and more interestingly, 2) over-expression of truncated and therefore aberrant 7SL RNA worsened this defect in DIS3L2 knockout cells as is marked by elevated expression of Six2, and IGF2, as well as calcium-sensing protein Camk1g (new Figure 6h). Thus, this data further underlines the essential function of DIS3L2 in regulation of renal cell differentiation in vitro and also helps connect the pathological role of truncated 7SL RNA to the defective differentiation of DIS3L2 deficient ESCs.

Reviewer #3 (Remarks to the Author):

Pirouz et al. analyze the consequences of removing Dis3l2 function in mammalian cells. Based on ribosome profiling, they report impaired translation of secretory and membrane proteins and claim a mechanistic link to impaired quality control of the 7SL RNA component of the signal recognition particle (SRP) that targets most proteins to the endoplasmic reticulum (ER) in mammalian cells. The authors try to link these findings to ER calcium leakage and dysregulated ESC differentiation. The experiments are well-executed and the independent observations of (i) different translation efficiencies, (ii) aberrant SRP 7SL RNA, (iii) calcium dysregulation, and (iv) differentiation phenotypes are clear. These results should be useful towards understanding the role of Dis3l2-mediated RNA decay in physiology. However, the suggested causality between Dis3l2 function, protein synthesis at the ER, and the proposed downstream “SRPopathy” phenotypes are weak and overinterpreted in this

manuscript as written. At minimum, the text implying these links should be significantly toned down before publication.

We thank this reviewer for their interest in our manuscript. We have performed several different experiments and analyses to further strengthen our conclusions and address these constructive comments:

Major comments:

1. Although the data showing aberrant SRP 7SL RNA in Dis3L2 KO cells (Fig. 3) are clear, the evidence linking Dis3L2 function and impaired “ER-mediated translation” is not convincing and requires more controls and/or reinterpretation:

a. In Fig. 1, if predicted ER-destined proteins make up at most 20% of translationally downregulated genes in KO cells (Fig. 1d), can a difference in ribosome occupancy by metagene analysis of the entire set of downregulated genes (Fig. 1b) be linked specifically to ER-associated translation? Do downregulated non-ER transcripts show the same RPF pattern in the KO cells? How do downregulated non-ER transcripts compared to unaffected non-ER transcripts? More results are needed to demonstrate that Dis3L2 KO is not inducing a more general translational repression mechanism.

We wish to clarify that the analysis in revised Figure 1e (previously Figure 1d) shows the percentage of ALL translated mRNAs and therefore (depending on the database) about 30-45% of all translated mRNAs are linked to the ER, and of these there is a strong and statistically significant decrease in translation efficiency (compared to up- or unchanged) using all 3 databases, with between ~30-50% of these (ER-linked) mRNAs being downregulated at the translation level in DIS3L2 deficient cells. Each of these 3 databases (TMHMM, SignalP, and Phobius) cannot themselves accurately predict all of the ER-associated transcripts, but rather each database considers different features to predict mRNAs that are translated at the ER. Nevertheless, according to this new analysis, ER-associated transcripts are significantly more enriched among translationally downregulated transcripts. Here (now Figure 1d in the revised manuscript) we show the distribution of RFPs in the translationally downregulated mRNAs. This distribution shows that while there is a decrease in RFPs throughout the body of the mRNA, there is an accumulation of RFPs at the 5' end that likely represents a stalling of Ribosomes. This distribution is clearly distinct from that of mRNAs with increased or unchanged translation efficiency (Supplemental Figure 1b).

Regarding whether or not DIS3L2 knockout induces a more general translational repression: Our genome-wide Ribo-Seq data as well as assessing cytosolic vs. ER-mediated translation using specific luciferase reporters clearly show that translational repression is not general and is restricted to ER-associated transcripts. Also, the metabolic labeling experiments show no change in bulk translation but rather a specific defect in translation of the secretome (Figures 2a, b). It is also interesting to note that mitochondrial related transcripts even show elevated translation (new Figure 1c), which merits further investigation to determine whether this is due to a direct or indirect effect of DIS3L2 loss. Furthermore, in the revised manuscript (new Figure 2f) we provide more detailed evidence supporting a defective ER-mediated translation in DIS3L2 knockout cells: using the dual luciferase reporter system we generated vectors containing firefly control and Renilla luciferases with or without endogenous ER-specific signal peptide (from insulin mRNA) and ER-retention signal, KDEL. Assessing luciferase activities of these and the parental reporters in the lysates and the supernatant samples obtained from control and DIS3L2 knockout mESCs, we now show that defective protein translation in DIS3L2 knockout cells depends on the presence of the signal peptide-coding sequence at the 5'-end of the luciferase reporter. Also removal of the KDEL signal from the 3'-end of the reporter directs luciferase reporters more effectively to the supernatant and diminishes their detection in the lysates, which again underlines the robustness and sensitivity of our reporter systems (new Figure 2f). These results clearly show that the inhibited protein translation upon DIS3L2 loss is specific to ER-associated transcripts. It is worth-mentioning that in these studies we used luciferase reporter systems containing the signal peptide directing the translated protein for secretion. However, ER-mediated translation generates proteins to be destined not only for secretion but also those to be sorted to other organelles particularly to reside in cellular membrane(s). In fact, translationally downregulated proteins in DIS3L2 knockout cells are enriched more greatly and more significantly among those predicted by “TMHMM” (please see Figure 1e) representing proteins containing transmembrane domains. Although designing and quantitatively assessing luciferase reporters for transmembrane proteins is more challenging, we believe that our reporter systems containing signal peptides sufficiently support the suppressed ER-mediated translation in the absence of DIS3L2.

b. Is there a Dis3L2-independent difference of ribosomal occupancy on ER-destined transcripts vs. others? What happens if the same analysis shown in Supplemental Fig. 1c is done for control cells?

We apologize for the confusion but “Supplemental Fig. 1c” was mislabeled. In fact, it represented RPF distribution of ER-associated transcripts in both control and knockout cells as this comment asks for. Please also refer to our response above.

c. Fig. 1d shows some preference of predicted ER proteins in the translationally downregulated genes, but these are still only ~1/8 of all predicted ER-targeted proteins. What makes these clients particularly sensitive to Dis3L2 KO? Are SRP clients de-enriched in the translationally up-regulated genes in the KO cells?

As it was mentioned in response to comment 1a, this is a misunderstanding and we apologize for not being completely clear in the original manuscript. This data is now broken down into down-, unchanged, and up-regulated transcripts (now presented as Revised Figure 1e). For detailed explanation please see response above. We furthermore discuss in the revised manuscript why some but not all ER-associated transcripts are affected by DIS3L2 loss. Briefly, 1) according to previously well-documented findings, not all the ER-associated transcripts utilize SRP-mediated mechanism(s) for their translation; and 2) a small fraction but not all of 7SL RNAs is misprocessed, uridylated, and targeted by DIS3L2; 3) a subset of mRNAs might be especially sensitive to SRP perturbation, which altogether could explain why not all the ER-mediated translation is blocked upon DIS3L2 loss. These points are now discussed in the revised manuscript.

d. I am not convinced that the metabolic labeling experiment in Fig. 2a is a fair assessment of secreted vs. non-secreted protein synthesis.

Apparently there were several misunderstandings regarding this experiment and we apologize for not being completely clear in the original manuscript. We have a new Supplementary Figure 2a depicting the experimental design to help clarify this.

It seems that the lysate and supernatant fractions were collected in separate experiments.

The lysate and supernatant fractions were not collected in separate experiments. In fact, they were collected from identical wells of the culture dishes. We have also repeated this experiment and added a later time point (New Figure 2a).

In addition, while the protein content of the lysates was normalized before scintillation counting, that of the supernatant was not addressed, allowing for far more variability that may arise not only from translation and secretion efficiency, but also different rates of cell growth and amino acid metabolism.

As the supernatants were collected from the same well of cultured cells from which lysates were prepared, and normalized to the total protein levels of the corresponding lysate samples, this rules out the possibility of “different rates of cell growth” and “amino acid metabolism” contributing to the observed results.

It's also not clear if the radioactivity measured in the supernatant is truly incorporated into proteins vs. free radiolabeled methionine. A more equal comparison would be to start with the same number of WT and KO cells, collect the supernatant and lysate in the same experiment, and remove free amino acids from each sample.

Indeed this is exactly how the experiment was performed. For more clarity to the reviewer and also to the readers, we now provide a schematic representation of the experimental details (please see Supplementary Figure 2a). The radioactivity measured in the supernatant is truly incorporated into proteins as free radiolabeled methionine (hot medium) was washed out before the first timepoint measurement. This was clear from the original manuscript as the measured radioactivity at “0 min” (right after the washing out hot medium and adding fresh cold medium) was close to zero. For all the experiments in this manuscript, an equal number of control and DIS3L2-deficient cells were plated before the experiments, and we believe the experiment scheme provided in Figure S2a should help resolve these questions.

e. Many experiments examining ER-mediated translation are done with a secreted luciferase reporter. To rule out SRP-independent effects, controls should be included showing the level of mRNA expression of this reporter in the conditions analyzed in Fig. 2c-2e, especially since the secreted and cytosolic reporters are in different plasmid backbones, encode different enzymes, and are expressed by transient transfection, which is additional subject to variable transfection efficiencies and cell-to-cell heterogeneity.

This is a critical question and we thank this reviewer for raising it. We had tested the mRNA expression of different luciferase reporter for the main part of the manuscript. Now the qRT-PCR data showing comparable expression levels of transfected reporters are provided in Supplementary Figure 2b-e and Supplementary Figure 5b,e. Moreover, in the revised manuscript (Figure 2f), we used new dual luciferase reporter systems (expressing

reporters from the same plasmids) to provide data supporting defective ER-mediated translation in Dis3l2 depleted cell (please also see response to comments 1a and 1f).

f. Stronger support for a link between Dis3l2 function and SRP client translation could be achieved by biochemically analyzing endogenous Dis3l2-dependent and independent clients by metabolic labeling and immunoprecipitations in control and KO cells with and without Dis3l2 re-expression. Alternatively, the authors could use matched reporter constructs that differ in the presence of an ER-targeting signal sequence, or compared to an SRP-independent ER targeting signal, such as a C-terminal tail-anchored transmembrane anchor.

This is a valid question and in response to this comment we engineered dual luciferase reporter systems, containing Renilla luciferases with or without ER-specific signal peptide (from the endogenous insulin mRNA sequence) and ER-retention signal, KDEL. Firefly Luciferase is used as a control. Assessing luciferase activities of these and the parental reporters in the lysates and the supernatant samples obtained from control and DIS3L2 knockout mESCs, we now show that defective protein translation in DIS3L2 knockout cells depends on the presence of the signal peptide-coding sequence at the 5'-end of the luciferase reporter. Also removal of the KDEL signal from the 3'-end of the reporter direct luciferase reporters more effectively to the supernatant and diminishes their detection in the lysates, again highlighting the robustness and sensitivity of our reporter systems (new Figure 2f). These results clearly show that the inhibited protein translation upon DIS3L2 loss is specific to ER-associated transcripts. It is worth-mentioning that in our entire studies we used luciferase reporter systems containing the signal peptide directing translated protein for secretion. However, ER-mediated translation generates proteins to be destined not only for secretion, but also those to be sorted to other organelles particularly to reside in the cellular membrane(s). In fact, translationally downregulated proteins in Dis3l2 knockout cells are enriched more greatly and more significantly among those predicted by "TMHMM" (please see Figure 1e) representing proteins containing transmembrane domains. Although designing and quantitatively assessing luciferase reporters for transmembrane proteins is more challenging, we believe that our reporter systems containing signal peptides sufficiently support the suppressed ER-mediated translation in the absence of DIS3L2.

To address defective ER-translation for endogenous mRNAs in the absence of DIS3L2, we used Min6 cells as an *in vitro* model of pancreatic beta-cells. Min6 cells sense increased glucose levels and metabolize glucose, which in turn leads to increased ATP/ADP level that triggers closure of a potassium channel. This causes cell membrane depolarization, calcium ion influx, and finally insulin secretion. Upon transient DIS3L2 depletion, ER-mediated translation (Fig. 4f) as well as glucose-stimulated insulin secretion (Fig. 4g) were attenuated in Min6 cells. Moreover, upon bypassing glucose sensing steps in Min6 cells by KCl administration, again DIS3L2-depleted cells failed to effectively secrete insulin (Fig. 4g). DIS3L2 knockdown in Min6 cells, caused ER calcium depletion, and increased cytosolic calcium (Fig. 4h) as well as increased calcium leakage from ER (Fig. 4i) similar to DIS3L2 knockout mESCs, suggesting a common defect in the regulation of calcium homeostasis. Finally, in contrast to control cells, DIS3L2-depleted Min6 cells failed to increase cytosolic calcium level upon glucose stimulation (Fig. 4j). This highlights the requirement of DIS3L2 for endogenous protein secretion, at least partially through its role in ER-mediated mRNA translation to prevent calcium leakage from ER.

g. Can the authors comment on the mechanism by which impaired SRP function in Dis3l2 KO cells leads to reduced translation, rather than protein mislocalization as observed in Costa et al. (2018), ref. 39?

Costa et al., (2018) used "the plant-based auxin-induced degradation (AID) system to induce acute and robust loss of one of the protein subunits of SRP, Srp72" in yeast. Utilizing this approach they observed that the synthesis of ER-associated proteins was perturbed and also some ER-destined proteins were mistargeted to mitochondria and caused mitochondrial dysfunction. This is a very interesting observation and we think this might also occur in DIS3L2 depleted cells since Gene Ontology of translationally upregulated mRNAs are highly enriched in mitochondria (Revised Figure 1c). This certainly merits more systematic investigation to address whether 1) ER-destined proteins are in fact mistargeted to mitochondria in DIS3L2 deficient cells, and 2) this is a direct or indirect consequence of DIS3L2 loss. The exact mechanism of how the accumulation of truncated/extended and uridylylated 7SL might inhibit SRP function remains to be determined and arguably this level of understanding is beyond the scope of this initial manuscript but our data support a defect resulting in ribosome stalling at the 5' end of signal-peptide containing mRNAs. This is discussed in the revised text.

2. The authors suggest a causal link between translational downregulation of ER-destined proteins and calcium leakage. While the data independently showing translational downregulation of some genes and calcium

dysregulation in Dis3l2 cells are clear, there is no evidence supporting the claim that the first causes the second. As an example, thapsigargin treatment causes cytosolic calcium accumulation, which also leads to impaired protein secretion in Dis3l2 KO cells (Fig. 2b and 2c). This already suggests that the two observations of impaired protein secretion/translation and calcium dysregulation cannot be cleanly uncoupled in these experiments.

We agree and thank the reviewer for raising this point. Accordingly, we changed the text in order to emphasize interrelationship between ER-associated mRNA translation and calcium homeostasis in the revised manuscript.

3. Similarly, while the experiments looking at cellular differentiation are beautiful, they remain observational. Descriptions suggesting causal links should be avoided in the text. Alternatively, the hypothesis should be directly tested. For example, rescuing the differentiation phenotype by re-expressing wildtype SRP 7SL RNA would give more credence to an “SRPopathy” model. As it stands, there is no data in the paper to rule out other plausible hypotheses. Perhaps Dis3l2 KO have misregulated miRNA targets that lead to the observed phenotypes. It also remains unclear why only a minority of predicted SRP clients are affected in an “SRPopathy” (see point 1b).

In the revised manuscript, we have provided more compelling evidence supporting the pathological effects of aberrant 7SL RNA accumulation on renal differentiation. Stable over-expression of truncated 7SL RNA in DIS3L2 knockout cells induces an even greater increase in the expression of renal progenitor markers Six2 and IGF2, as well as calcium sensing factor Camk1g (Figure 6h). This effect was not obvious in control cells most likely due to the presence of DIS3L2 protein and its quality control function on truncated 7SL RNA. These results underscore our “SRPopathy” model. However, in the revised manuscript, we have toned down on causal effect of aberrant 7SL RNA processing and prudently discussed alternative mechanisms.

Minor points:

1. The authors demonstrate their expertise in RNA biology and metabolism in the first paragraph of the introduction. In contrast, the second paragraph describing SRP function was vague, lacking specific information that could influence interpretations in the manuscript. The general mechanism of SRP-mediated targeting of ribosomes synthesizing secretory and membrane proteins to the ER, as well as the scope and characteristics of SRP clients, are well studied and included in most college-level biology textbooks, such as Molecular Biology of the Cell by Alberts et al. It would be helpful if the relevant parts of this established work are introduced to help place the authors' findings in context.

The related text is revised now and hopefully provides a better background for the readers.

2. The term/concept of “ER-mediated translation” is not clear to me. It implies that ribosomes at the ER translate more efficiently. Is this what the authors mean (and if so, what is the evidence for this)? Or does it refer only to protein synthesis at the ER?

We refer to protein synthesis at the ER. We have changed to ‘ER-mediated mRNA translation’ or protein synthesis at the ER’ throughout the text.

3. Please be wary of the term “protein translation”. mRNAs are translated, proteins are not. Alternatives are “translation” or “protein synthesis”.

This is now corrected in the revised manuscript.

4. “aberrant” is misspelled as “abberant” throughout the text.

This is now corrected in the revised manuscript.

REVIEWERS' COMMENTS:

Reviewer #1 (Remarks to the Author):

In their revised manuscript, Pirouz and colleagues have adequately addressed all concerns that I had after reading the initial version of the paper. Specifically, they verified histone mRNA expression in mESCs, showing that steady-state levels of these transcripts are not significantly deregulated in this particular cell type, as opposed to other cell lines. Moreover, the authors have now eliminated several technical shortcomings pointed out in the referee's report. They also provided explanations which rationalize the use of scintillation instead of phosphorimaging in the analysis of secreted newly synthesized proteins. Additional experiments performed during revision argue against the link between rRNA or tRNA misprocessing caused by DIS3L2 knock-out and altered mRNA translation. Instead, data from experiments performed using reporters convincingly support authors' conclusions pointing to 7SL RNA quality control deficiency and SRP malfunction as the major cause of associated molecular phenotypes. Although experiments with Osr1-GFP reporter in DIS3L2 knock-out vs wt cells have not been done, the authors' attempts to further demonstrate the utility of shRNA-mediated approach have to be appreciated. Furthermore, new data for knock-out cells are now provided to demonstrate up-regulation of renal-specific markers and synergistic effect of overexpression of truncated 7SL RNA specific to these cells, but not observed in the wt control. Together with additional experiments carried out in response to other reviewers' reports, amendments and clarifications in the text, the manuscript by Pirouz et al. has gained a lot of quality and I recommend accepting it for publication in Nature Communications without further hesitation.

Reviewer #2 (Remarks to the Author):

The authors have generally done a good job of responding to reviewer comments. However, I still have a hard time accepting the central theme of the paper that the accumulation of uridylated 7SL is the underlying mechanism connecting the loss of DIS3L2 with the various downstream phenotypes (defects in ER associated mRNA translation, ER Ca²⁺ retention, and stem cell differentiation). The reason is that the fraction of uridylated 7SL relative to non-uridylated 7SL appears to be remarkably low (as shown in their new figure 3F). It is unclear to me how this very small fraction could have such a large downstream impact. In addition, they show that they can rescue the defect by overexpressing full-length 7SL suggesting some sort of mass effect (i.e. reversible competition between uridylated and non-uridylated 7SL) (fig. 3i). However, the overexpression increases total 7SL by less than 10% (supplementary fig. 5d). I cannot see how this very slight increase in normal 7SL can reverse the effect of what was a very small fraction of uridylated 7SL. As for figures 3d,e,g, they show a reasonably large fold increase of uridylated 7SL in the various pull downs when comparing mutant to wt, but given that uridylated 7SL is extremely low to non-detectable in the wt, the fold increase is likely to reflect an extremely low absolute value, again making it hard to understand how it could be causing such severe downstream phenotypes. I appreciate that this may be a very difficult question to resolve, but it does represent a central theme of their paper. Maybe it just requires some finessing of their conclusions or maybe I am just missing

something. If other reviewers disagree with my interpretation, I am happy to acquiesce. There are a lot of nice data in the paper.

More minor issues: I remain unconvinced that the difference between ctrl and ko in figure 1d is significant and truly represents a block/delay in ribosome progression on transcripts in DIS3L knockouts. Also, now that they include a up group in their updated figure 1e, there appears to be an equal or near-equal enrichment of ER-translated proteins in the down and up subsets (KO/Ctrl) when considering either the SignalP or Phobius ontology groups (new figure 1e). If DIS3L is supposedly inhibiting translation of ER-translated proteins, why are they also enriched in the up group.

Reviewer #3 (Remarks to the Author):

The authors have sufficiently addressed my main concerns in their revised manuscript. I support publication in Nature Communications, with a few minor points regarding the preciseness of the writing and descriptions of what occurs when treating cells with certain small molecules that can influence readers' interpretations of the results.

Minor Points:

1. In the explanation of Fig. 2: thapsigargin does not selectively inhibit translation of ER-targeted mRNAs. It inhibits the SERCA channel at the ER, which activates the integrated stress response (via the phosphorylation of eIF2-alpha), which inhibits protein synthesis globally (again, not specifically of ER-targeted substrates) while upregulating stress response proteins.
2. There is a similar issue with the explanation of the use of anisomycin in Fig. 4: anisomycin is primarily known to act as a translation elongation inhibitor. It directly binds ribosomes to inhibit peptide bond formation. The inhibition of calcium leak at ER translocons is a secondary effect from the inability in cells to clear ribosomes from translocons, but this is not clearly explained (and the citations to references 45 and 54 do not sufficiently support a primary function of anisomycin in preventing calcium leakage)
2. I am still struggling with the term "ER-mediated translation". It is not a term used in the long-standing fields of ER translocation or protein synthesis, and appears to only refer to translation that occurs at the ER membrane. However, the ER is not actively facilitating translation itself so the description seems somewhat misleading. Alternatives might be "ER-targeted" or "ER-localized" translation.

REVIEWERS' COMMENTS:

Reviewer #1 (Remarks to the Author):

In their revised manuscript, Pirouz and colleagues have adequately addressed all concerns that I had after reading the initial version of the paper. Specifically, they verified histone mRNA expression in mESCs, showing that steady-state levels of these transcripts are not significantly deregulated in this particular cell type, as opposed to other cell lines. Moreover, the authors have now eliminated several technical shortcomings pointed out in the referee's report. They also provided explanations which rationalize the use of scintillation instead of phosphorimaging in the analysis of secreted newly synthesized proteins. Additional experiments performed during revision argue against the link between rRNA or tRNA misprocessing caused by DIS3L2 knock-out and altered mRNA translation. Instead, data from experiments performed using reporters convincingly support authors' conclusions pointing to 7SL RNA quality control deficiency and SRP malfunction as the major cause of associated molecular phenotypes. Although experiments with Osr1-GFP reporter in DIS3L2 knock-out vs wt cells have not been done, the authors' attempts to further demonstrate the utility of shRNA-mediated approach have to be appreciated. Furthermore, new data for knock-out cells are now provided to demonstrate up-regulation of renal-specific markers and synergistic effect of overexpression of truncated 7SL RNA specific to these cells, but not observed in the wt control. Together with additional experiments carried out in response to other reviewers' reports, amendments and clarifications in the text, the manuscript by Pirouz et al. has gained a lot of quality and I recommend accepting it for publication in Nature Communications without further hesitation.

We highly appreciate the reviewer's comments and support they provide for our work.

Reviewer #2 (Remarks to the Author):

The authors have generally done a good job of responding to reviewer comments. However, I still have a hard time accepting the central theme of the paper that the accumulation of uridylated 7SL is the underlying mechanism connecting the loss of DIS3L2 with the various downstream phenotypes (defects in ER associated mRNA translation, ER Ca²⁺ retention, and stem cell differentiation). The reason is that the fraction of uridylated 7SL relative to non-uridylated 7SL appears to be remarkably low (as shown in their new figure 3F). It is unclear to me how this very small fraction could have such a large downstream impact. In addition, they show that they can rescue the defect by overexpressing full-length 7SL suggesting some sort of mass effect (i.e. reversible competition between uridylated and non-uridylated 7SL) (fig. 3i). However, the overexpression increases total 7SL by less than 10% (supplementary fig. 5d). I cannot see how this very slight increase in normal 7SL can reverse the effect of what was a very small fraction of uridylated 7SL. As for figures 3d,e,g, they show a reasonably large fold increase of uridylated 7SL in the various pull downs when comparing mutant to wt, but given that uridylated 7SL is extremely low to non-detectable in the wt, the fold increase is likely to reflect an extremely low absolute value, again making it hard to understand how it could be causing such severe downstream phenotypes. I appreciate that this may be a very difficult question to resolve, but it does represent a central theme of their paper. Maybe it just requires some finessing of their conclusions or maybe I am just missing something. If other reviewers disagree with my interpretation, I am happy to acquiesce. There are a lot of nice data in the paper.

We are grateful to the reviewer for raising an interesting point of discussion. We acknowledge that the level of mis-processed 7SL RNA is low in the absence of DIS3L2. However, it is notable that correspondingly, the level of ER-targeted mRNA translation is also not entirely abrogated in DIS3L2-depleted cells (only ~20-30% reduction). This suggests that relative minor dysregulation of 7SL RNA processing can result in changes in ER-targeted mRNA translation and ER function. We have nevertheless toned down our interpretations as suggested by the reviewer in our revised text.

More minor issues: I remain unconvinced that the difference between ctrl and ko in figure 1d is significant and truly represents a block/delay in ribosome progression on transcripts in DIS3L knockouts. Also, now that they include a up group in their updated figure 1e, there appears to be an equal or near-equal enrichment of ER-translated proteins in the down and up subsets (KO/Ctrl) when considering either the SignalP or Phobius ontology groups (new figure 1e). If DIS3L is supposedly inhibiting translation of ER-translated proteins, why are they also enriched in the up group.

We wish to clarify that upregulated genes (unlike the downregulated subset) are not significantly enriched in ER-translated proteins, as it is shown in Figure 1e. We have discussed the difference between different ontology groups in the revised text and the previous response to the reviewer. Also we used elaborate molecular and biochemical approaches to address specific downregulation of ER-targeted mRNA translation in the absence of DIS3L2.

Reviewer #3 (Remarks to the Author):

The authors have sufficiently addressed my main concerns in their revised manuscript. I support publication in Nature Communications, with a few minor points regarding the preciseness of the writing and descriptions of what occurs when treating cells with certain small molecules that can influence readers' interpretations of the results.

Minor Points:

1. In the explanation of Fig. 2: thapsigargin does not selectively inhibit translation of ER-targeted mRNAs. It inhibits the SERCA channel at the ER, which activates the integrated stress response (via the phosphorylation of eIF2-alpha), which inhibits protein synthesis globally (again, not specifically of ER-targeted substrates) while upregulating stress response proteins.

We thank reviewer for this helpful comment. We have addressed this point in the revised manuscript text.

2. There is a similar issue with the explanation of the use of anisomycin in Fig. 4: anisomycin is primarily known to act as a translation elongation inhibitor. It directly binds ribosomes to inhibit peptide bond formation. The inhibition of calcium leak at ER translocons is a secondary effect from the inability in cells to clear ribosomes from translocons, but this is not clearly explained (and the citations to references 45 and 54 do not sufficiently support a primary function of anisomycin in preventing calcium leakage).

We thank the reviewer for this comment. In the revised manuscript we explain the anisomycin effect more accurately.

3. I am still struggling with the term "ER-mediated translation". It is not a term used in the long-standing fields of ER translocation or protein synthesis, and appears to only refer to translation that occurs at the ER membrane. However, the ER is not actively facilitating translation itself so the description seems somewhat misleading. Alternatives might be "ER-targeted" or "ER-localized" translation.

We have changed the phrase "ER-mediated translation" to "ER-targeted translation" throughout the revised text.